# Understanding Generalized Label Smoothing when Learning with Noisy Labels

## Abstract

Label smoothing (LS) is an arising learning paradigm that uses the positively weighted average of both the hard training labels and uniformly distributed soft labels. It was shown that LS serves as a regularizer for training data with hard labels and therefore improves the generalization of the model. Later it was reported LS even helps with improving robustness when learning with noisy labels. However, we observe that the advantage of LS vanishes when we operate in a high label noise regime. Puzzled by the observation, we proceeded to discover that several proposed learning-with-noisy-labels solutions in the literature instead relate more closely to *negative label smoothing* (NLS), which defines as using a negative weight to combine the hard and soft labels! We show that NLS differs substantially from LS in their achieved model confidence. To differentiate the two cases, we will call LS the positive label smoothing (PLS), and this paper unifies PLS and NLS into *generalized label smoothing* (GLS). We provide understandings for the properties of GLS when learning with noisy labels. Among other established properties, we theoretically show NLS is considered more beneficial when the label noise rates are high. We provide extensive experimental results on multiple benchmarks to support our findings too.

## 1 Introduction

Label smoothing (LS) (Szegedy et al., 2016) is an arising learning paradigm that uses positively weighted average of both the hard training labels and uniformly distributed soft label:

$$\mathbf{y}^{\text{LS},r} = (1 - r) \cdot \mathbf{y} + \frac{r}{K} \cdot \mathbf{1} \tag{1}$$

where we denote the one-hot vector form of hard label and an all one vector as $\mathbf{y}$, $\mathbf{1}$ respectively. $K$ is the number of label classes, and $r$ is the smooth rate in the range of $[0, 1]$. It was shown that LS serves as a regularizer for the hard training data and therefore improves generalization of the model. The regularizer role of LS prevents the model from fitting overly on the target class. Empirical studies have demonstrated the effectiveness of LS in improving the model performance across various benchmarks (Pereyra et al., 2017) (such as image classification (Szegedy et al., 2016), machine translation (Vaswani et al., 2017), language modelling (Chorowski & Jaitly, 2017)) and model calibration (Müller et al., 2019).

Later it was reported LS even helps with improving robustness when learning with noisy labels (Lukasik et al., 2020). However, we observe that the advantage of LS vanishes when we operate in a high label noise regime. In Figure 1, we present a set of experiments on UCI datasets (Dua & Graff, 2017). We highlight best two smooth rates (possible to have tied smooth rates) under each label noise rate. Indeed, non-negative smooth rates (circles colored in red) outperform negative ones when the label noise rates are low. Nonetheless, with the increasing of noise rates, negative smooth rates $r < 0$ (Eqn. (1), diamonds colored in green) appear to be more competitive when learning with noisy labels. Puzzled by the observation, we proceeded to discover

that several proposed learning-with-noisy-labels solutions in the literature, including Loss Correction (Patrini et al., 2017), NLNL (Kim et al., 2019) and Peer Loss (Liu & Guo, 2020), instead relate more closely to *negative label smoothing* (NLS), which defines as using a negative weight to combine the hard and soft labels!

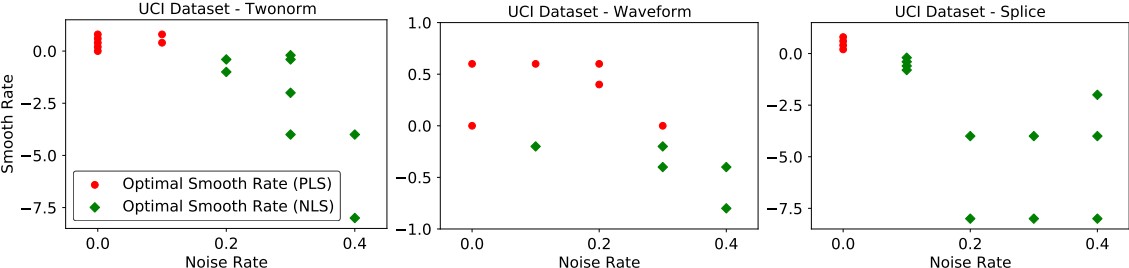

Figure 1: Optimal smooth rates on UCI datasets with different label noise rates.

This paper unifies label smoothing with either positive or negative smooth rate into a *generalized label smoothing* (GLS) framework. Our paper is motivated by the above inconsistent observations. And we aim to provide a more thorough understanding of GLS under the setting of learning with noisy labels, rather than proposing a new method to compete for state-of-the-art performances. We firstly show that negative label smoothing differs substantially from positive label smoothing in their achieved model confidence. With the presence of label noise, we then proceed to show that there exists a phase transition behavior when finding the optimal label smoothing rate for GLS. Particularly, when label noise is low, positive label smoothing is able to uncover the optimal model while negative label smoothing is considered more beneficial in a high label noise regime.

We provide extensive experimental evidences to support our findings. For instance, on multiple benchmark datasets, we present the clear transition of the optimal smoothing rate going from positive to negative when we keep increasing noise rates. On CIFAR-10 test data, we show a negative smoothing rate elicits higher model confidence on correct predictions and lower confidence on wrong predictions compared with the behavior of a positive one.

Our contributions summarize as follows:

- We provide understandings for a generalized notion of label smoothing (GLS) when learning with noisy labels, where the label smooth rate can go negative. (An understanding paper rather than a method paper)

- We show that several robust loss functions in the noise learning literature correspond to learning with GLS, under certain noise rate models. (Section 3)

- We show that negative label smoothing improves the confidence of model prediction. With the presence of label noise, we demonstrate learning with a negative smooth rate can be more robust to label noise compared with a positive rate when label noise rates are high. (Section 4, 5)

- Empirical experiments on multiple datasets and methods demonstrate that with the presence of label noise, NLS becomes competitively robust to label noise. We also empirically show how GLS results in trade-offs in model confidence, bias and variance of the generalization error. (Section 6, Appendix D and E)

- Extensive experiment results validate our main theoretical conclusions. Besides, we also discuss practical considerations and multi-class extensions of GLS to mitigate the impact of label noise. (Appendix B, C)

We defer all proofs to Appendix F. Our work primarily contributes to the literature of learning with noisy labels (Scott et al., 2013; Natarajan et al., 2013; Liu & Tao, 2015; Patrini et al., 2017; Liu & Guo, 2020). Our core results are contingent on recent works of understanding the effect of label smoothing when training deep neural network models, i.e., label smoothing improves model calibration (Müller et al., 2019), more complexed forms of label smoothing (Li et al., 2020; Yuan et al., 2020), and in particular when label noise

presents (Lukasik et al., 2020; Liu, 2021). We generalize a concept called generalized label smoothing and provide new understandings for when, instead of setting a positive smoothing rate as the literature would normally do, a negative smoothing rate is considered a better option. Due to the space limit, we defer a more detailed discussion of related works to Appendix A.

## 2 PRELIMINARIES

### 2.1 LEARNING WITH SMOOTHED LABELS

For a $K$-class classification task, we denote by $X \in \mathcal{X}$ a high-dimensional feature and $Y \in \mathcal{Y} := \{1, 2, ..., K\}$ the corresponding label. Suppose $(X, Y) \in \mathcal{X} \times \mathcal{Y}$ are drawn from a joint distribution $\mathcal{D}$. Let $\mathbf{y_i}$ be the one-hot encoded vector form of $y_i$ which generates according to $Y$. The random variable of smoothed label $Y^{\mathrm{LS},r}$ with smooth rate $r \in [0, 1]$ generates $\mathbf{y_i^{\mathrm{LS},r}}$ as (Szegedy et al., 2016):

$$\mathbf{y_i^{\mathrm{LS},r}} = (1 - r) \cdot \mathbf{y_i} + \frac{r}{K} \cdot \mathbf{1}$$

For example, when $r = 0.3$, the smoothed label of $\mathbf{y_i} = [0, 1, 0]$ becomes $\mathbf{y_i^{\mathrm{LS},r=0.3}} = [0.1, 0.8, 0.1]$.

We consider a broader setting where the smoothed label might be negatively related to the corresponding feature. As a supplementary to existed works on label smoothing, we explore the benefits of learning with generalized label smoothing (GLS), i.e., $r \in (-\infty, 1]$ instead of a non-negative $r$.

$$\mathbf{y_i^{\mathrm{GLS},r}} := (1 - r) \cdot \mathbf{y_i} + \frac{r}{K} \cdot \mathbf{1} \tag{2}$$

where $\mathbf{y_i^{\mathrm{GLS},r}}$ is given by the random variable of generalized smooth label $Y^{\mathrm{GLS},r}$. We name the scenario $r < 0$ as negative label smoothing (NLS). To clarify, we don't assume a strict lower bound for $r$. If $r \to -\infty$, normalizing $\mathbf{y_i^{\mathrm{GLS},r}}$ by $1 - r$ returns $\mathbf{y_i^{\mathrm{LS},r}} = \mathbf{y_i} - \frac{\mathbf{1}}{K}$. We will show when imposing a negative smoothing parameter will be considered beneficial as compared to a positive one. In the main paper, we focus on the binary classification task where $y_i \in \{0, 1\}$ and $K = 2$. And we defer multi-class extensions to Appendix C. Denote $f$ as a deep neural network, $\mathbf{f}(\mathbf{x_i})$ is the model prediction of $x_i \in X$ with element $\mathbf{f}(\mathbf{x_i})_{y_i} := \mathbb{P}(Y = y_i | X = x_i, f)$. Given the sample $x \in \mathcal{X}$ and a hard label $y \in \mathcal{Y}$, binary CE loss is defined as $\ell_{\mathrm{CE}}(\mathbf{f}(\mathbf{x}), y) := -\log(\mathbf{f}(\mathbf{x})_y)$. Throughout this paper, we shorthand $\ell$ as $\ell_{\mathrm{CE}}$ for a clean presentation.

### 2.2 LEARNING WITH NOISY LABELS

The noisy label literature considers the setting where we only have access to samples with noisy labels from $(X, \widetilde{Y})$. Suppose random variables $(X, \widetilde{Y}) \in \mathcal{X} \times \widetilde{\mathcal{Y}}$ are drawn from a noisy joint distribution $\widetilde{\mathcal{D}}$. Statistically, the random variable of noisy labels $\widetilde{Y}$ can be characterized by a noise transition matrix $T$, where each element $T_{i,j}$ represents the probability of flipping the clean label $Y = i$ to the noisy label $\widetilde{Y} = j$, i.e, $T_{ij} = \mathbb{P}(\widetilde{Y} = j | Y = i)$. In this paper, we are interested in the widely studied class-dependent label noise. We assume the label noise is conditionally independent of features, i.e.,

$$\mathbb{P}(\widetilde{Y} = j | Y = i) = \mathbb{P}(\widetilde{Y} = j | X, Y = i), \forall i, j \in [K]$$

For the binary classification setting, define $e_0 := \mathbb{P}(\widetilde{Y} = 1 | Y = 0)$, $e_1 := \mathbb{P}(\widetilde{Y} = 0 | Y = 1)$. Without loss of generality, we assume $e_1 - e_0 = e_\Delta \geq 0$. We denote the binary noise transition matrix in the noisy label setting as: $T = \begin{pmatrix} 1 - e_0 & e_0 \\ e_1 & 1 - e_1 \end{pmatrix}$.

## 2.3 MODEL CONFIDENCE

We define a key quantity, model confidence, that plays an important role in later sections.

**Definition 1.** *Model confidence of model $f$ for sample $(x, y)$. Given a model $f$, a sample $x$ with its target label $y \in \{0, 1\}$, the model confidence of $f$ w.r.t. sample $x$ is defined as $MC(f; x, y) = \mathbf{f}(\mathbf{x})_y - \mathbf{f}(\mathbf{x})_{1-y}$.*

$MC(f; x, y)$ in definition 1 characterizes the difference of the predicted probability between target class and the other class. $MC(f; x, y) = 0$ simply means $f$ has no confident on its predictions since the model can not identify the target class of $x$. $MC(f; x, y)$ returns a negative value when $f$ gives a wrong prediction and is not confident to predict the label of $x$ as the target label $y$. To dig into how GLS influences the model confidence on correct and wrong predictions in following sections, we separate the distribution $\mathcal{D}$ into

$$\mathcal{D}_f^+ := \{(X, Y) \sim \mathcal{D} : MC(f; X, Y) > 0\}, \quad \mathcal{D}_f^- := \{(X, Y) \sim \mathcal{D} : MC(f; X, Y) \leq 0\}$$

## 3 CONNECTION TO OTHER ROBUST METHODS

In this section, we aim to theoretically explore the connection between GLS and popular methods such as backward/forward loss correction (Natarajan et al., 2013; Patrini et al., 2017), NLNL (Kim et al., 2019) and peer loss (Liu & Guo, 2020). We defer the corresponding empirical validations to Appendix B.

For $r \leq 1$, let $\tilde{\mathbf{y}}$ be the vector form of noisy label $\tilde{y}$ obtained from $\widetilde{Y}$, we define the $r$ smoothed label of $\tilde{y}$ as $\tilde{\mathbf{y}}^{\text{GLS},r}$, where $\tilde{\mathbf{y}}^{\text{GLS},r} := (1 - r) \cdot \tilde{\mathbf{y}} + (r/K) \cdot \mathbf{1}$ and is generated by the random variable $\widetilde{Y}^{\text{GLS},r}$. Risk minimization of the Generalized Label Smoothing (GLS) w.r.t. noisy labels becomes:

$$\text{Risk Minimization Using GLS:} \quad \min \mathbb{E}_{(X,\widetilde{Y}) \sim \widetilde{\mathcal{D}}}\left[\ell(\mathbf{f}(\mathbf{X}), \widetilde{Y}^{\text{GLS},r})\right] \quad (3)$$

The GLS framework covers three special methods: PLS ($r \in (0, 1]$), Vanilla (CE) Loss ($r = 0$) and NLS ($r < 0$). Besides, we observe that NLS connects to a special case of label smoothing regularization. We highlight this in Theorem 1.

**Theorem 1.** $\forall r \in [0, 1]$, *NLS with smooth rate $-r$ is a special form of label smoothing regularization:*

$$\min \mathbb{E}_{(X,\widetilde{Y}) \sim \widetilde{\mathcal{D}}}\left[\ell(\mathbf{f}(\mathbf{X}), \widetilde{Y}^{GLS,-r})\right] = \min \mathbb{E}_{(X,\widetilde{Y}) \sim \widetilde{\mathcal{D}}}\left[2 \cdot \ell(\mathbf{f}(\mathbf{X}), \widetilde{Y}) - \ell(\mathbf{f}(\mathbf{X}), \widetilde{Y}^{GLS,r})\right]$$

### 3.1 LOSS CORRECTION

Loss correction (Patrini et al., 2017) studies two robust loss designs which are based on the knowledge of non-singular noise transition matrix $T$. The backward correction $\ell^{\leftarrow}(\mathbf{f}(\mathbf{X}), \widetilde{Y})$ re-weights the loss $\ell(\mathbf{f}(\mathbf{X}), \widetilde{Y})$ by $T_{\hat{Y}, \widetilde{Y}}^{-1}$ with $\hat{Y}$ being the model predicted label, while the proposed forward correction $\ell^{\rightarrow}(\mathbf{f}(\mathbf{X}), \widetilde{Y})$ multiplies the model predictions by $T$.

**Proposition 1.** *For $r_{LC} := \frac{2e_0}{2e_0 - 1} < 0$, $\lambda_{LC} := e_\Delta \cdot \frac{1}{1 - 2e_0}$, risk minimization of both backward and forward correction (with the knowledge of noise rates) are equivalent to the combination of NLS and an extra bias term Bias-LC*

$$\min \mathbb{E}_{(X,\widetilde{Y}) \sim \widetilde{\mathcal{D}}}\left[\ell^{\leftarrow}(\mathbf{f}(\mathbf{X}), \widetilde{Y})\right] = \min \mathbb{E}_{(X,\widetilde{Y}) \sim \widetilde{\mathcal{D}}}\left[\ell^{\rightarrow}(\mathbf{f}(\mathbf{X}), \widetilde{Y})\right]$$

$$= \min \mathbb{E}_{(X,\widetilde{Y}) \sim \widetilde{\mathcal{D}}}\left[\ell(\mathbf{f}(\mathbf{X}), \widetilde{Y}^{GLS,r_{LC}})\right] + \lambda_{LC} \cdot \underbrace{\mathbb{E}_{X,Y=1}\left[\ell(\mathbf{f}(\mathbf{X}), 1) - \ell(\mathbf{f}(\mathbf{X}), 0)\right]}_{\textit{Bias-LC}}$$

The incurred Bias-LC controls the model confidence on $(X, Y = 1) \sim \mathcal{D}_f$. Note that when the noise rate is not substantially high, i.e, $e_0 \in [0, \frac{1}{2})$, $\lambda_{\mathrm{LC}} > 0$. Then, compared with loss correction, NLS with smooth rate $r_{\mathrm{LC}}$ makes the model $f$ to be less confident on $(X, Y = 1) \sim \mathcal{D}_f^+$ and more confident on $(X, Y = 1) \sim \mathcal{D}_f^-$ (wrong predictions). However, the impact of term Bias-LC is diminishing when either $e_\Delta \to 0$ (symmetric noise rates) or $e_0 \to 0$ (low noise rates) as specified in Theorem 2.

**Theorem 2.** *Assume the noise transition matrix is symmetric, i.e, $e_\Delta = 0$, backward and forward loss correction are a special form of NLS with smooth rate $r_{LC}$.*

### 3.2 LEARNING FROM COMPLEMENTARY LABELS

Complementary label (Ishida et al., 2017) was firstly introduced to mitigate the cost of collecting data. Rather than encouraging the model to fit directly on the target, learning from complementary labels trains the model to not fit on the complementary label which differs from the target. Later, an indirect training method "Negative Learnin" (NL) (Kim et al., 2019) was proposed to reduce the risk of providing incorrect information with the presence of noisy labels and is robust to label noise in multi-class classification tasks. A more generic unbiased risk estimator of learning with complementary labels was proposed (Ishida et al., 2019) and is defined as: $\ell_{\mathrm{CL}}(\mathbf{f}(\mathbf{X}), \widetilde{Y}) := \ell(\mathbf{f}(\mathbf{X}), \widetilde{Y}) - \ell(\mathbf{f}(\mathbf{X}), 1 - \widetilde{Y})$.

**Theorem 3.** *Learning from complementary labels with $\ell_{CL}$ is equivalent to NLS with smooth rate $r_{CL} \to -\infty$:*

$$\min \mathbb{E}_{(X, \widetilde{Y}) \sim \widetilde{\mathcal{D}}} \left[ \ell_{CL}(\mathbf{f}(\mathbf{X}), \widetilde{Y}) \right] \iff \min \mathbb{E}_{(X, \widetilde{Y}) \sim \widetilde{\mathcal{D}}} [\ell(\mathbf{f}(\mathbf{X}), \widetilde{Y}^{GLS, r_{CL} \to -\infty})]$$

### 3.3 PEER LOSS FUNCTIONS

Peer loss functions (Liu & Guo, 2020) propose a family of robust loss measures which do not require the knowledge of noise rates. The mathematical representation of peer loss functions is $\ell_{\mathrm{PL}}(\mathbf{f}(\mathbf{X}), \widetilde{Y}) := \ell(\mathbf{f}(\mathbf{X}), \widetilde{Y}) - \ell(\mathbf{f}(\mathbf{X_1}), \widetilde{Y}_2)$, where $(X_i, \widetilde{Y}_i) \sim \widetilde{\mathcal{D}}$. The second term of peer loss evaluates on randomly paired data samples and labels to punish $f$ from overly fitting on noisy labels.

**Proposition 2.** *For $r_{PL} := 2 \cdot \mathbb{P}(\widetilde{Y} = 1)$, $\lambda_{PL} := 1 - r_{PL}$, risk minimization of peer loss is equivalent to negative label smoothing regularization with an extra term Bias-PL, i.e.,*

$$\min \mathbb{E}_{(X, \widetilde{Y}) \sim \widetilde{\mathcal{D}}} \left[ \ell_{PL}(\mathbf{f}(\mathbf{X}), \widetilde{Y}) \right] = \min \mathbb{E}_{(X, \widetilde{Y}) \sim \widetilde{\mathcal{D}}} \left[ \ell(\mathbf{f}(\mathbf{X}), \widetilde{Y}) - \ell(\mathbf{f}(\mathbf{X}), \widetilde{Y}^{GLS, r_{PL}}) \right]$$

$$+ \lambda_{PL} \cdot \underbrace{\mathbb{E}_{X, \widetilde{Y} = 1} \left[ \ell(\mathbf{f}(\mathbf{X}), 1) - \ell(\mathbf{f}(\mathbf{X}), 0) \right]}_{\text{Bias-PL}}$$

The incurred term Bias-PL controls the model confidence on $(X, \widetilde{Y} = 1) \sim \widetilde{\mathcal{D}}$ and has a diminishing effect as $\mathbb{P}(\widetilde{Y} = 1) \to 1/2$. Generally, peer loss relates to GLS as the negatively weighted GLS term appears to be a regularizer. Note that we have access to the $\mathbb{P}(\widetilde{Y} = 1)$, we can bridge the gap between GLS and peer loss by adding an estimable term Bias-PL. With some derivations, we further show in Theorem 4, when noisy priors are equal, peer loss has an exact GLS form.

**Theorem 4.** *When the noisy labels have equal prior, i.e, $\mathbb{P}(\widetilde{Y} = 0) = \mathbb{P}(\widetilde{Y} = 1)$, peer loss is a special form of NLS regularization with smooth rate $r_{PL}$. Besides,*

$$\min \mathbb{E}_{(X, \widetilde{Y}) \sim \widetilde{\mathcal{D}}} \left[ \ell_{PL}(\mathbf{f}(\mathbf{X}), \widetilde{Y}) \right] \iff \min \mathbb{E}_{(X, \widetilde{Y}) \sim \widetilde{\mathcal{D}}} \left[ \ell(\mathbf{f}(\mathbf{X}), \widetilde{Y}^{GLS, r \to -\infty}) \right]$$

## 4 GLS AND MODEL CONFIDENCE

Now we show that NLS differs substantially from PLS in their achieved model confidence. This discussion sets the foundation for our discussion when learning with noisy labels in next section.

### 4.1 GLS ON CLEAN DATA

We firstly show that NLS differs substantially from PLS in their achieved model confidence. This discussion sets the foundation for our discussion when learning with noisy labels in proceeding subsections.

When the label is clean, i.e, $e_0 = e_1 = 0$, Eqn. (3) reduces to:

$$\min \mathbb{E}_{(X,Y)\sim\mathcal{D}}\Big[\ell(\mathbf{f}(\mathbf{X}),Y)\Big] + \frac{r}{2} \cdot \mathbb{E}_{(X,Y)\sim\mathcal{D}}\underbrace{\Big[\ell(\mathbf{f}(\mathbf{X}),1-Y) - \ell(\mathbf{f}(\mathbf{X}),Y)\Big]}_{\text{Term MC}_\ell(f;X,Y)} \tag{4}$$

To clarify, we are not restricting $\mathcal{D}$ to have infinite samples, i.e., for the discrete distribution $\mathcal{D} = \{x_i, y_i\}_{i=1}^N$, Eqn. (4) becomes: $\min \Big[\frac{1}{N}\sum_{i\in[N]}\ell(\mathbf{f}(\mathbf{x_i}),y_i)\Big] + \frac{r}{2N}\Big[\sum_{i\in[N]}\big(\ell(\mathbf{f}(\mathbf{x_i}),1-y_i) - \ell(\mathbf{f}(\mathbf{x_i}),y_i)\big)\Big]$. Note:

$$\text{MC}_\ell(f; X, Y) = \log\Big(\mathbf{f}(\mathbf{X})_Y / \big(1 - \mathbf{f}(\mathbf{X})_Y\big)\Big); \quad \text{MC}(f; X, Y) = 2 \cdot \mathbf{f}(\mathbf{X})_Y - 1$$

Both $\log\big(x/(1-x)\big)$ and $2x - 1$ are monotonically increasing for $x \in (0,1)$, model $f$ with a high $\text{MC}_\ell(f;X,Y)$ has high $\text{MC}(f;X,Y)$. The difference between PLS and NLS lie in the weight of Term $\text{MC}_\ell(f;X,Y)$: NLS encourages high $\text{MC}_\ell(f;X,Y)$ and $\text{MC}(f;X,Y)$ while PLS has an opposite effect.

### 4.2 INSIGHTS FROM AN EMPIRICAL OBSERVATION

We adopt the generation of 2D (binary) synthetic dataset from (Amid et al., 2019) by randomly sampling two circularly distributed classes. The inner annulus indicates one class (blue), while the outer annulus denotes the other class (red). We hold $20\%$ data samples for performance comparison.

**Side-effects of over-confident** In Figure 2, the colored bands depict the different levels of prediction probabilities: light blue + orange bands indicate samples that satisfy $\text{MC} < 0.4$ (low model confidence). When learning with clean data, GLS with non-positive smooth rate may yield over-confidence on the model prediction and a relatively low test accuracy.

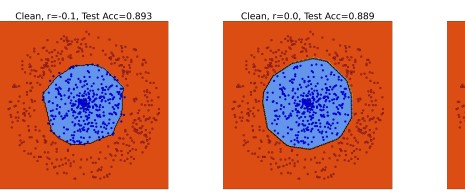

Figure 2: Model confidence visualization of GLS on synthetic data (Type 1) with the clean data. $r^* \in [0, 0.4]$. (left: NLS; middle: Vanilla Loss; right: PLS).

**Label noise reduces model confidence** Recent works (Liu, 2021; Cheng et al., 2020) have demonstrated that with the presence of label noise, learning with noisy labels directly will eventually result in unconfident model predictions. Continuing the synthetic 2D dataset, we flip the clean labels according to a symmetric noise transition matrix with noise rate $e_i$ for both classes. With the presence of label noise in Figure 3, GLS generally

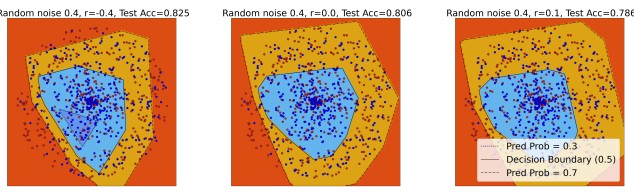

Figure 3: Model confidence visualization of GLS on synthetic data (Type 1) with noise rate $e_i = 0.4$. $r_{\text{opt}} = -0.4$. (left: NLS; middle: Vanilla Loss; right: PLS).

becomes less confident on its predictions. Besides, when the smooth rate increases from negative to positive, more samples are of uncertain predictions. Thus, a smaller/negative smooth rate is beneficial when the noise rate increases by encouraging more confident predictions.

## 5 GLS WITH LABEL NOISE

In this section, we target at the optimal candidates of $r$ in GLS when the label noise presents. In Section 4, we have shown that NLS and PLS have the opposite functionality on the model confidence when training on clean data. Given the unseen test data, learning with non-negative smooth rates may not always return the best outcome (Figure 3). Based on this observation, we delve into details to show when NLS is more favorable than PLS and Vanilla Loss. We start with stating Assumption 1:

**Assumption 1.** *We assume learning with clean data distribution $\mathcal{D}$ with smooth rate $r^* \leq 1$ in GLS returns the best performance on the unseen clean test data distribution $\mathcal{D}_{test}$.*

Assumption 1 simply offers an "anchor" point to initiate our analysis for the noisy label setting. To clarify, we don't rule out the possibility that other methods outperform GLS with optimal smooth rate $r^*$. Later in Section 6.1 and Appendix D, we will empirically test what $r^*$ usually is on various benchmarks. We define optimal classifier on $Y^*$ which follows $r^*$ smooth label distribution as: $f_{\mathcal{D}}^* := \arg\min_f \mathbb{E}_{(X,Y)\sim\mathcal{D}}\Big[\ell(\mathbf{f}(\mathbf{X}), Y^*)\Big]$. With the introduction of $r^*$ and $f_{\mathcal{D}}^*$, our goal is then to recover the classifier $f$ using the noisy training labels. To bridge learning with noisy labels and clean labels for GLS, we define $\lambda_1, \lambda_2$ and offer Theorem 5.

$$\lambda_1 := \Big[(e_0 - \frac{r^*}{2}) + (1 - 2e_0) \cdot \frac{r}{2}\Big], \quad \lambda_2 := e_\Delta \cdot (1 - r)$$

**Theorem 5.** *The risk minimization of GLS (Eqn. (3)) in the noisy setting relates to the risk defined on the clean data with two additional bias terms:*

$$\min \quad \underbrace{\mathbb{E}_{(X,Y)\sim\mathcal{D}}\Big[\ell(\mathbf{f}(\mathbf{X}), Y^*)\Big]}_{True\ Risk} + \underbrace{\lambda_1 \cdot \mathbb{E}_{(X,Y)\sim\mathcal{D}}\Big[\ell(\mathbf{f}(\mathbf{X}), 1 - Y) - \ell(\mathbf{f}(\mathbf{X}), Y)\Big]}_{M\text{-}Inc1}$$
$$+ \underbrace{\lambda_2 \cdot \mathbb{E}_{X,Y=1}\Big[\ell(\mathbf{f}(\mathbf{X}), 0) - \ell(\mathbf{f}(\mathbf{X}), 1)\Big]}_{M\text{-}Inc2} \quad (5)$$

The True Risk is the risk minimization w.r.t. clean optimal label distribution $Y^*$. Training GLS on noisy labels results in two extra bias terms which affect the model confidence. We defer an empirical validation of Theorem 5 to Appendix B. Now we proceed to answer "what parameters are preferred in the noisy setting".

### 5.1 SYMMETRIC ERROR RATES WITH $e_\Delta = 0$

Symmetric error rates $e := e_0 = e_1$ indicates the probability of flipping to the other class is equal for both classes. In this case, $\lambda_2 = 0$ and Term M-Inc2 is cancelled and Eqn. (5) reduces to

$$\min \quad \underbrace{\mathbb{E}_{(X,Y)\sim\mathcal{D}}\Big[\ell(\mathbf{f}(\mathbf{X}), Y^*)\Big]}_{True\ Risk} + \underbrace{\lambda_1 \cdot \mathbb{E}_{(X,Y)\sim\mathcal{D}}\Big[\ell(\mathbf{f}(\mathbf{X}), 1 - Y) - \ell(\mathbf{f}(\mathbf{X}), Y)\Big]}_{M\text{-}Inc1} \quad (6)$$

**Noisy labels impairs model confidence on Vanilla Loss** In the GLS framework, define the optimal $r$ that will cancel the impact of Term M-Inc1 as:

$$\text{when } r_{\text{opt}} := \frac{r^* - 2e}{1 - 2e}, \quad \text{M-Inc1} = 0 \quad (7)$$

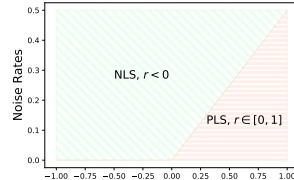

The threshold $r_{\text{opt}}$ in Eqn. 7 implies:

**Theorem 6.** *With Assumption 1, GLS with smooth rate $r = r_{opt}$ yields $f_{\mathcal{D}}^*$.*

• *When error rate $e < r^*/2$, $r = r_{opt} > 0$ (PLS);*

Figure 4: Decision between NLS, PLS given $e, r^*$.

- *When error rate $e = r^*/2$, $r = 0$ (Vanilla Loss);*
- *When error rate $e > r^*/2$, $r = r_{opt} < 0$ (NLS).*

In Theorem 6, adopting NLS when noise rate $e < r^*/2$ induces $\lambda_1 < 0$, Term M-Inc1 makes $f$ overly-confident on its predictions compared with $Y^*$. In Figure 4, with the decreasing of $r^*$, PLS is less tolerant of labels with high noise. Similarly, if $e \geq \frac{r^*}{2}$, with the decreasing of $r^*$, NLS is more robust in the high noise regime while PLS makes the model $f$ become less-confident on its predictions. Clearly, NLS outperforms PLS especially when noise rates are large and $r^*$ is small.

## 5.2 ASYMMETRIC ERROR RATES WITH $e_\Delta \neq 0$

In this case, adopting $r = \frac{r^* - 2e_0}{1 - 2e_0}$ removes the Term M-Inc1. However, when $r < 1$, Term M-Inc2 is not negligible due to assymetric noise transition matrix. As a result, Term M-Inc2 becomes:

$$e_\Delta \cdot \frac{1 - r^*}{1 - 2e_0} \cdot \mathbb{E}_{X, Y=1}\Big[\ell(\mathbf{f}(\mathbf{X}), 0) - \ell(\mathbf{f}(\mathbf{X}), 1)\Big], \quad \text{with } e_\Delta \cdot \frac{1 - r^*}{1 - 2e_0} \geq 0$$

Term M-Inc2 in the minimization increases the model confidence on $(X, Y = 0) \sim \mathcal{D}_f^+$. The model will then become overly-confident with the class that has a low noise rate $e_0$. Meanwhile, Term M-Inc2 decreases the model confidence on $(X, Y = 1) \sim \mathcal{D}_f^+$ (less-confident to the class with a high noise rate $e_1$).

## 6 EXPERIMENT RESULTS

In this section, we present our empirical observations regarding the role of GLS under clean and noisy labels by using UCI datasets, CIFAR-10 and CIFAR-100.

### 6.1 WHAT IS THE PRACTICAL DISTRIBUTION OF $r^*$ AND $r_{\text{OPT}}$?

$r^*$ **and $r_{\text{opt}}$ on UCI datasets (Dua & Graff, 2017)**   As for UCI datasets, we pick Twonorm and Splice for illustration in the main paper. The noisy labels are generated by a symmetric noise transition matrix with noise rate $e_i = [0.1, 0.2, 0.3, 0.4]$. As highlighted in Table 1, $r_{\text{opt}}$ appears with positive values when the data is clean (same as $r^*$) or of a low noise rate. With the increasing of noise rates, the performance of PLS results in a much larger degradation compared with NLS. We color-code different noise regimes where either PLS (red-ish) or NLS (green-ish) outperforms the other. Clearly there is a separation of the favored smoothing rate for different noise scenarios (upper left & low noise for PLS, bottom right & high noise for NLS).

Table 1: Test accuracies of GLS on clean and noisy UCI datasets with best two smooth rates (green: NLS; red: PLS). Results on more benchmark datasets are deferred to Appendix D.

| Smooth Rate | Twonorm | | | | | Splice | | | | |
|---|---|---|---|---|---|---|---|---|---|---|
| | $e_i = 0$ | $e_i = 0.1$ | $e_i = 0.2$ | $e_i = 0.3$ | $e_i = 0.4$ | $e_i = 0$ | $e_i = 0.1$ | $e_i = 0.2$ | $e_i = 0.3$ | $e_i = 0.4$ |
| $r = 0.8$ | 0.990 | 0.990 | 0.986 | 0.982 | 0.968 | 0.980 | 0.946 | 0.919 | 0.856 | 0.760 |
| $r = 0.6$ | 0.990 | 0.989 | 0.987 | 0.981 | 0.972 | 0.978 | 0.939 | 0.913 | 0.869 | 0.778 |
| $r = 0.4$ | 0.990 | 0.990 | 0.987 | 0.983 | 0.971 | 0.978 | 0.948 | 0.922 | 0.885 | 0.797 |
| $r = 0.2$ | 0.990 | 0.989 | 0.986 | 0.986 | 0.969 | 0.978 | 0.948 | 0.919 | 0.878 | 0.800 |
| $r = 0.0$ | 0.990 | 0.989 | 0.987 | 0.985 | 0.973 | 0.976 | 0.948 | 0.926 | 0.876 | 0.806 |
| $r = -0.4$ | 0.986 | 0.988 | 0.988 | 0.986 | 0.972 | 0.961 | 0.956 | 0.928 | 0.880 | 0.817 |
| $r = -0.6$ | 0.986 | 0.988 | 0.987 | 0.984 | 0.974 | 0.961 | 0.956 | 0.926 | 0.880 | 0.819 |
| $r = -1.0$ | 0.986 | 0.986 | 0.988 | 0.985 | 0.977 | 0.956 | 0.954 | 0.932 | 0.889 | 0.819 |
| $r = -2.0$ | 0.986 | 0.986 | 0.986 | 0.986 | 0.978 | 0.952 | 0.946 | 0.935 | 0.898 | 0.830 |
| $r = -4.0$ | 0.986 | 0.986 | 0.986 | 0.986 | 0.983 | 0.946 | 0.943 | 0.939 | 0.911 | 0.830 |
| $r = -8.0$ | 0.986 | 0.986 | 0.986 | 0.985 | 0.986 | 0.943 | 0.946 | 0.939 | 0.915 | 0.845 |
| $r_{\text{opt}} =$ | [0.0, 0.8] | [0.4, 0.8] | [-1.0, -0.4] | [-4.0, -0.4] | -8.0 | [0.0, 0.8] | [-0.6, -0.4] | [-8.0, -4.0] | -8.0 | -8.0 |

$r^*$ **and $r_{\text{opt}}$ on CIFAR datasets (Krizhevsky et al., 2009)**   When learning with a larger scale and more complex dataset, like CIFAR-10 and CIFAR-100, models are prone to converge on a local optimal solution

rather than the global optimum. This phenomenon occurs frequently in NLS which ends up with performance degradation. Thus, in Table 2, when learning with noisy labels, we report the better performance of GLS between direct training and loading the same warm-up model. And we observe that the performance of NLS is more competitive than PLS when learning with clean data. Clearly, NLS outperforms PLS in CIFAR-10 and CIFAR-100 under various synthetic noise settings. The gap is larger when the noise rates are high.

Table 2: Test accuracies (meanstd) of GLS on synthetic noisy CIFAR datasets. Best two smooth rates for each synthetic noise setting are highlighted for each $\epsilon$ (green: NLS; red: PLS).

| Smooth Rate | CIFAR-10 Symmetric | | | | CIFAR-10 Asymmetric | | CIFAR-100 Symmetric | |
|---|---|---|---|---|---|---|---|---|
| | $\varepsilon = 0.0$ | $\varepsilon = 0.2$ | $\varepsilon = 0.4$ | $\varepsilon = 0.6$ | $\varepsilon = 0.2$ | $\varepsilon = 0.3$ | $\varepsilon = 0.4$ | $\varepsilon = 0.6$ |
| $r = 0.8$ | 92.91±0.06 | 88.88±1.61 | 81.48±2.91 | 73.16±0.16 | 90.45±0.06 | 87.83±0.13 | 54.04±0.93 | 39.50±0.18 |
| $r = 0.6$ | 92.33±0.09 | 87.50±1.31 | 82.11±0.86 | 73.59±0.15 | 90.41±0.09 | 87.83±0.13 | 52.72±0.15 | 40.49±0.07 |
| $r = 0.4$ | 93.05±0.04 | 87.13±0.07 | 81.50±1.42 | 74.21±0.19 | 90.49±0.10 | 87.90±0.13 | 54.26±0.07 | 41.57±0.05 |
| $r = 0.0$ | 91.44±0.16 | 85.08±0.86 | 80.42±2.29 | 75.34±0.13 | 88.32±0.24 | 86.27±0.32 | 48.03±0.29 | 38.11±0.14 |
| $r = -0.4$ | 93.55±0.06 | 87.55±0.08 | 81.58±0.19 | 75.95±0.13 | 87.27±1.83 | 88.33±0.06 | 56.87±0.08 | 43.70±0.16 |
| $r = -0.8$ | 92.74±0.05 | 88.46±0.11 | 81.56±0.15 | 76.15±0.14 | 86.40±1.32 | 87.96±0.43 | 57.35±0.08 | 44.10±0.06 |
| $r = -1.0$ | 92.58±0.08 | 88.58±0.08 | 81.95±0.10 | 76.20±0.10 | 88.47±0.15 | 87.50±0.73 | 57.44±0.09 | 43.85±0.19 |
| $r = -2.0$ | 93.30±0.03 | 88.78±0.09 | 83.64±0.15 | 76.11±0.07 | 88.66±0.17 | 87.27±0.70 | 58.10±0.08 | 44.88±0.11 |
| $r = -4.0$ | 93.13±0.04 | 88.90±0.07 | 84.34±0.13 | 77.22±0.09 | 89.56±0.17 | 87.29±0.59 | 58.35±0.09 | 46.38±0.05 |
| $r = -6.0$ | 93.14±0.08 | 88.94±0.11 | 84.52±0.13 | 77.42±0.16 | 89.70±0.24 | 87.57±0.42 | 57.73±0.10 | 46.46±0.09 |

## 6.2 COMPARISONS BETWEEN GLS AND MORE ROBUST METHODS

In Table 3, we compare GLS with several robust methods in synthetic noisy CIFAR datasets. We emphasize that we are not proposing a new method to compete with state-of-the-art methods. Instead, we hope to help readers understand how the generalized label smoothing fare when label noise presents. Clearly, GLS (especially NLS) can definitely be viewed as a competitive and efficient robust loss function which outperforms Cross Entropy, Bootsrap (Reed et al., 2014), SCE (Wang et al., 2019), APL (Ma et al., 2020) and Forward correction (Patrini et al., 2017) in most settings.

Table 3: Performance comparisons on synthetic noisy CIFAR datasets: we adopt the same model architecture for all methods (ResNet 34 (He et al., 2016)), best achieved test accuracy is reported.

| Method | CIFAR-10, Symmetric | | | CIFAR-10, Asymmetric | | CIFAR-100, Symmetric | |
|---|---|---|---|---|---|---|---|
| | $\varepsilon = 0.2$ | $\varepsilon = 0.4$ | $\varepsilon = 0.6$ | $\varepsilon = 0.2$ | $\varepsilon = 0.3$ | $\varepsilon = 0.4$ | $\varepsilon = 0.6$ |
| Cross Entropy | 86.45 | 82.72 | 74.04 | 88.59 | 86.14 | 48.20 | 38.27 |
| Bootstrap (Reed et al., 2014) | 86.06 | 81.65 | 75.26 | 87.69 | 85.51 | 47.28 | 35.81 |
| Forward correction (Patrini et al., 2017) | 84.85 | 84.98 | 73.97 | 89.42 | 88.25 | 53.04 | 41.59 |
| SCE (Wang et al., 2019) | 89.39 | 80.31 | 75.28 | 88.07 | 85.93 | 49.34 | 38.87 |
| APL (Ma et al., 2020) | 88.42 | 81.27 | 76.62 | 88.75 | 87.41 | 51.63 | 42.31 |
| Peer Loss (Liu & Guo, 2020) | 90.21 | 86.40 | 79.64 | 91.38 | 89.65 | 62.16 | 53.72 |
| ELR (Liu et al., 2020) | 92.57 | 91.32 | 88.86 | 93.48 | 92.21 | 68.03 | 60.49 |
| AUM (Pleiss et al., 2020) | 91.52 | 87.85 | 81.71 | 92.17 | 90.63 | 59.29 | 44.05 |
| Positive Label Smoothing (PLS) | 90.24 | 83.78 | 75.01 | 90.61 | 88.04 | 55.17 | 41.63 |
| Negative Label Smoothing (NLS) | 89.05 | 84.85 | 77.82 | 90.02 | 88.42 | 58.47 | 46.58 |

**Additional results** More extensive empirical results are included in Appendix: empirical validation of main theorems (Appendix B); practical considerations of GLS (Appendix C); experiment results on additional datasets (Appendix D); bias and variance of the generalization error on the clean data (Appendix E).

**Conclusion** In this paper, we provide understandings for a generalized notion of label smoothing where the label smoothing rate can go negative. We show that learning with negatively smoothed labels explicitly improves the confidence of model prediction. This key property acts as a significant role when the confidence of model prediction drops. We make connections between negative label smoothing and existing learning with noisy label solutions. In contrast to existing works that promote the use of positive label smoothing, we show both theoretically and empirically the advantage of a negative smooth rate when the label noise rate increases. Our observations provide new understanding for the effects of label smoothing, especially when the training labels are imperfect.

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

# APPENDIX

The Appendix is organized as follows.

- Section A presents the full version of related works.
- Section B includes empirical validations of theoretical conclusions in Section 3 and Theorem 5.
- Section C discusses practical considerations and multi-class extensions of GLS.
- Section D shows additional experiments on synthetic dataset and UCI datasets.
- Section E illustrates the bias and variance trade-off when learning with GLS from clean data.
- Section F includes omitted proofs for theoretical conclusions in the main paper.

## A    FULL VERSION OF RELATED WORKS

Our work supplements to two lines of related works.

**Learning with noisy labels**    Annotated labels from human labelers usually consists of an non-negligible amount of mis-labeled data samples. Making deep neural nets perform robust training on "noisily" labeled datasets remains a challenge. Classical approaches of learning with noisy labels assume the noisy labels are independent to features. They firstly estimate the noise transition matrix (Liu & Tao, 2015; Menon et al., 2015; Harish et al., 2016; Patrini et al., 2017), then proceed with a loss correction (Natarajan et al., 2013; Patrini et al., 2017; Liu & Tao, 2015) to mitigate label noise. Recent works propose robust loss functions (Xu et al., 2019; Kim et al., 2019; Liu & Guo, 2020; Wei & Liu, 2020) to train deep neural nets directly without the knowledge of noise rates, or design a pipeline which dynamically select and train on "clean" samples with small loss (Jiang et al., 2018; Han et al., 2018; Yu et al., 2019; Yao et al., 2020a). More recently, several approaches target at addressing more challenging noise settings, such as instance-dependent label noise (Cheng et al., 2020; Berthon et al., 2021).

**Understanding the effect of label smoothing**    Learning with one-hot labels is prone to over-fitting, soft label learning then naturally draws attentions of machine learning researchers. Successful applications of soft label learning include the label distribution learning (Geng, 2016) which provides an instance with description degrees of all the labels. Label smoothing (LS) (Szegedy et al., 2016) is another arising learning paradigm that uses positively weighted average of both the hard training labels and uniformly distributed soft labels. Empirical studies have demonstrated the effectiveness of LS in improving the model performance (Pereyra et al., 2017; Szegedy et al., 2016; Vaswani et al., 2017; Chorowski & Jaitly, 2017) and model calibration (Müller et al., 2019). However, knowledge distilling a teacher network (trained on smoothed labels) into a student network is much less effective (Müller et al., 2019). Later, generalization effects of more advanced forms of label smoothing was studied, such as structural label smoothing (Li et al., 2020) and non-uniform label smoothing (Chen et al., 2020). More recently, it was shown that an appropriate label smoothing regularizer with reduced label variance boosts the convergence (Xu et al., 2020). When label noise presents, (Liu, 2021) gives theoretical justifications for the memorizing effects of label smoothing. And the effectiveness of label smoothing in mitigating label noise is investigated in (Lukasik et al., 2020).

## B    EMPIRICAL VALIDATIONS OF MAIN THEOREMS

In this section, we empirically validate our main theoretical conclusions, i.e, the connection between GLS and popular methods, and the comparison between clean empirical risks and noisy ones.

## B.1 CONNECTION TO OTHER ROBUST METHODS

We firstly compare GLS with backward correction (Natarajan et al., 2013), forward correction (Patrini et al., 2017) and peer loss (Liu & Guo, 2020) on CIFAR-10 dataset. To approximate the performance of backward/forward Loss Correction, we adopt GLS with smooth rate $\frac{\epsilon}{(\epsilon-1)}$. As for the approximation of peer loss, we choose $\ell(\mathbf{f}(\mathbf{X}), \widetilde{Y}) - \ell(\mathbf{f}(\mathbf{X}), \widetilde{Y}^{\text{GLS},r=0.5})$ which is equivalent to GLS when $r \to -\infty$. Experiment results in Table 4 on CIFAR-10 under symmetric noise settings demonstrate that the equivalent forms of GLS are robust to label noise.

Table 4: Comparison of test accuracies on CIFAR-10 under symmetric label noise.

| Method | CIFAR-10, Symmetric | | |
|---|---|---|---|
| | $\varepsilon = 0.2$ | $\varepsilon = 0.4$ | $\varepsilon = 0.6$ |
| Backward $T$ (Patrini et al., 2017) | 84.79 | 83.40 | 71.52 |
| Forward $T$ (Patrini et al., 2017) | 84.85 | **83.98** | 73.97 |
| GLS form | **87.33** | 81.73 | **75.80** |
| Peer Loss (Liu & Guo, 2020) | **90.21** | **86.40** | **79.64** |
| GLS form | 88.98 | 85.05 | 76.66 |

**Explanation of the performance gap** In practice, we adopt the same hyper-parameter setting as used for all other smooth rates for GLS form. Loss corrections will firstly warm-up with the cross-entropy loss, estimate the noise transition matrix with this pre-trained model, and then proceed to train with the backward/forward corrected loss. Peer loss functions adopt a dynamical adjustment for learning rate. The warming up, estimation error of noise transition matrix as well as the special hyper-parameter settings explain performance gaps.

## B.2 CLEAN EMPIRICAL RISK V.S. NOISY EMPIRICAL RISK

Now we empirically verify Theorem 8, which relates the risk of GLS in the noisy setting to the clean ones. Assume the the noise label is generated through the symmetric noise transition matrix. We restate the connection between two risks as below:

$$\underbrace{\mathbb{E}_{(X,\widetilde{Y})\sim\widetilde{\mathcal{D}}}\Big[\ell(\mathbf{f}(\mathbf{X}), \widetilde{Y}^{\text{GLS},r})\Big]}_{\textbf{Noisy Risk}} = \underbrace{\mathbb{E}_{(X,Y)\sim\mathcal{D}}\Big[\ell(\mathbf{f}(\mathbf{X}), Y^*)\Big]}_{\textbf{(Clean) True Risk}} + \underbrace{\lambda_1 \cdot \mathbb{E}_{(X,Y)\sim\mathcal{D}}\Big[\ell(\mathbf{f}(\mathbf{X}), 1-Y) - \ell(\mathbf{f}(\mathbf{X}), Y)\Big]}_{\textbf{Gap of Risk}}$$

In practice, obtaining the exact value of $r^*$ is not trivial. Finding an accurate $r$ for GLS to make $\lambda_1$ be exactly 0 is pretty hard. Note that when M-Inc1 is equipped with the Cross-entropy loss and NLS encourages confident predictions, one mistake could make the value of M-Inc1 has a huge difference. To be more specific, suppose the label of an image is a "cat", if the NLS-trained model makes a mistake on this image, it is very likely to predict this image "cat" with a probability that $p \to 0$ since NLS may push the model become overly-confident. Then $\log(p)$ can be extremely small. Thus, the empirical risk (evaluated on the clean data) for NLS will change much more significantly than PLS ones.

We use a UCI dataset (Waveform) for illustration. The value of $r^*$ is approximately 0, and the estimated (clean) true risk in above is 0.1798. When the noise rates are $0.1, 0.2, 0.3, 0.4$, the optimal smooth rate should be $-0.25, -0.67, -1.5, -4$ according to Eqn. (7). The estimated noisy risk of GLS on these noise settings can be summarized in Table 1. Clearly, when $e = 0.1$, $r = -0.25$ is closest to the estimated clean true risk (also returns the best test accuracy among these smooth rates). Similarly observations reach to all other $e$. Although the risks evaluated on noisy settings with the corresponding $r_{\text{opt}}$ can not be exactly the same as the true risk (evaluates on the clean data with $r = 0.0$), $r_{\text{opt}}$ already reaches the smallest one among these smooth rates and the non-zero **Gap of Risk** is best explained by the precision of $r^*$ as well as the estimation error of risks.

Table 5: The difference between the empirical true risk of $Y^*$ on the clean data and empirical risk GLS on noisy labels (UCI-Waveform dataset): $r^*$, empirical true risk, and empirical noisy risks under various noise levels are highlighted in purple.

| Smooth rate | Risk (clean) | Risk ($e = 0.1$) | Risk ($e = 0.2$) | Risk ($e = 0.3$) | Risk ($e = 0.4$) |
|---|---|---|---|---|---|
| $r = 0.8$ | 0.6773 | 0.6831 | 0.6873 | 0.6899 | 0.6923 |
| $r = 0.6$ | 0.6295 | 0.6521 | 0.6689 | 0.6833 | 0.6905 |
| $r = 0.4$ | 0.5437 | 0.5994 | 0.6408 | 0.6718 | 0.6873 |
| $r = 0.2$ | 0.4134 | 0.5212 | 0.5956 | 0.6550 | 0.6828 |
| $\mathbf{r^* = 0.0}$ | **0.1798** | 0.4057 | 0.5399 | 0.6314 | 0.6758 |
| $r = -0.25$ | -36.8095 | **0.1983** | 0.4381 | 0.5957 | 0.6685 |
| $r = -0.67$ | -333.1283 | -28.3508 | **0.2167** | 0.5132 | 0.6503 |
| $r = -1.5$ | -97.4378 | -61892.8047 | -94.9509 | **0.1911** | 0.6003 |
| $r = -4.0$ | -270.6456 | -179.0037 | -93.1257 | -156.4539 | **0.1165** |

## C    PRACTICAL CONSIDERATION OF GLS

In the main paper, we theoretically show when we should adopt NLS and PLS. In this section, we discuss more practical considerations of GLS, including the optimal smoothing parameter, how to reduce the impacts of bias terms, and multi-class extensions.

### C.1    THE OPTIMAL SMOOTHING PARAMETER

In practice, we don't have access to noise rates $e_i$. Our work does not intend to particularly focus on the noise rate estimation. For readers interested in the noise rate estimation, please refer to (Liu & Tao, 2015; Menon et al., 2015; Harish et al., 2016; Patrini et al., 2017; Yao et al., 2020b; Zhu et al., 2021). To estimate $r_{\mathrm{opt}} = \frac{r^* - 2e}{1 - 2e}$, one can simply assume $r^* \to 0$. And the noise rate $e$ is estimable by a large family of noise estimation methods mentioned above. Our practical observations show that NLS with a CE warm-up is not sensitive to the negative smooth rate, for example, on CIFAR-10 and CIFAR-100 synthetic noisy datasets, $r < -1.0$ frequently achieves best results (see Table 2 in the main paper). Our current contribution focuses on understanding the generalized label smoothing, and we prefer leaving the task of identifying the optimal smooth rate to future works.

### C.2    MAKING GLS MORE ROBUST TO LABEL NOISE

There is a line of related works targeting at distinguishing clean labels from the noisy labels. Current literature in selecting clean samples from noisily labeled dataset is based on the empirical evidence that samples with noisy/wrong labels have a larger loss than clean ones. For interested readers, please refer to (Han et al., 2018; Jiang et al., 2018; Yu et al., 2019; Yao et al., 2020a; Wei et al., 2020; Northcutt et al., 2021). Compared with the risk minimization over the clean data distribution $(X, Y) \sim \mathcal{D}$, learning directly with GLS on the noisy distribution $(X, \widetilde{Y}) \sim \widetilde{\mathcal{D}}$ will result in an extra term $(e_1 - e_0) \cdot (1 - r) \cdot \mathbb{E}_{(X, Y=1) \sim \mathcal{D}}[\ell(\mathbf{f}(\mathbf{X}), 0) - \ell(\mathbf{f}(\mathbf{X}), 1)]$ compared to the clean scenario. Empirically, we can estimate the bias term, perform a bias correction by subtracting the estimated bias term from the objective function in Eqn. (3).

Suppose we have access to a clean distribution $\mathcal{D}_{\mathrm{clean}}$ which consists of selected clean samples. Denote the estimated noise rates as $\hat{e}_i$, when $e_\Delta \neq 0$, in order to make GLS be more robust to label noise and fit on the optimal distribution $Y^*$, we improve GLS by performing a model confidence correction on the dominating class through:

$$\text{GLS-C:} \quad \min \; \mathbb{E}_{(X,\widetilde{Y})\sim\widetilde{\mathcal{D}}} \left[ \ell\big(\mathbf{f}(\mathbf{X}), \widetilde{Y}^{\text{GLS},r}\big) \right]$$
$$-(\hat{e}_1 - \hat{e}_0) \cdot (1-r) \cdot \mathbb{E}_{(X,Y=1)\sim\mathcal{D}_{\text{clean}}} \underbrace{\left[ \ell\big(\mathbf{f}(\mathbf{X}), 0\big) - \ell\big(\mathbf{f}(\mathbf{X}), 1\big) \right]}_{\text{confidence correction}}$$

### C.3 MULTI-CLASS EXTENSION

As an extension to the binary classification task, we next show how GLS extends to the multi-class setting under two broad families of noise transition model. When learning with multi-class classification tasks, we assume Assumption 1 holds in the multi-class setting. And for $Y, \widetilde{Y} \in [K]$, we extend the definition of model confidence to multi-class classification tasks as:

**Definition 2.** *Model confidence of sample $x$ (K-class classification). Given a model $f$, a sample $x$ with its target label $y \in [K]$, the model confidence score of $f$ w.r.t. sample $x$ is defined as*

$$MC(f; x, y) = \mathbf{f}(\mathbf{x})_y - \frac{1}{K-1} \sum_{i \neq y} \mathbf{f}(\mathbf{x})_i$$

**Sparse noise transition matrix** Sparse noise model (Wei & Liu, 2020) assumes $K$ is an even number. For $c \in [K/2]$, $i_c < j_c$, sparse noise model specifies $K/2$ disjoint pairs of classes $(i_c, j_c)$ to simulate the scenario where particular pairs of classes are ambiguity and misleading for human annotators. The off-diagonal element of $T$ reads $T_{i_c, j_c} = e_0$, $T_{j_c, i_c} = e_1$. Suppose $e_0 + e_1 < 1$, the diagonal entries become $T_{i_c, i_c} = 1 - e_1$, $T_{j_c, j_c} = 1 - e_0$. Clearly, our conclusions in Section 5 extends directly to the sparse noise transition matrix by simply splitting the $K$-class classification task into $\frac{K}{2}$ disjoint binary ones.

**Symmetric noise transition matrix** Symmetric noise model (Kim et al., 2019) is a widely accepted synthetic noise model in the literature of learning with noisy labels. The symmetric noise model generates the noisy labels by randomly flipping the clean label to the other possible classes with probability $\epsilon$. $\forall i \neq j$, $T_{i,j} = \epsilon/(K-1)$, and the diagonal entry $T_{i,i} = 1 - \epsilon$. Define the optimal $r$ for GLS in the multi-class setting as $r_{\text{opt}} := \frac{(K-1)\cdot r^* - K \cdot \epsilon}{(K-1) - K \cdot \epsilon}$, Theorem 6 can be extended to the multi-class setting as:

**Theorem 7.** *Under Assumption 1, suppose the symmetric noise rate is not too large, i.e, $\epsilon < \frac{K-1}{K}$, GLS with smooth rate $r = r_{opt}$ yields $f_{\mathcal{D}}^*$.*

- *When error rate $\epsilon < \frac{(K-1)\cdot r^*}{K}$, $r = r_{opt} > 0$ (PLS);*
- *When error rate $\epsilon = \frac{(K-1)\cdot r^*}{K}$, $r = 0$ (Vanilla Loss);*
- *When error rate $\epsilon > \frac{(K-1)\cdot r^*}{K}$, $r = r_{opt} < 0$ (NLS).*

## D ADDITIONAL EXPERIMENT RESULTS AND DETAILS

In this section, we include more experiment results, observations and details for learning with GLS.

### D.1 EXPERIMENT DETAILS ON CIFAR-10, CIFAR-100

We firstly introduce experiment details on CIFAR-10 dataset adopted in our experiment designs.

**Training settings of clean CIFAR-10 dataset (Krizhevsky et al., 2009)**    We adopted ResNet34 (He et al., 2016), trained for 200 epochs with batch-size 128, SGD (Robbins & Monro, 1951) optimizer with Nesterov momentum of 0.9 and weight decay 1e-4. The learning rate of first 100 epochs is 0.1. Then it multiples with 0.1 for every 50 epochs.

**Generating noise labels on CIFAR datasets**    We adopt symmetric noise model which generates noisy labels by randomly flipping the clean label to the other possible classes with probability $\epsilon$. And we set $\epsilon = 0.2, 0.4, 0.6$ for CIFAR-10, $\epsilon = 0.4, 0.6$ for CIFAR-100. We also make use of asymmetric noise model. The asymmetric noise is generated by flipping the true label to the next class with probability $\epsilon$. We set $\epsilon = 0.2, 0.3$ for CIFAR-10.

**Training settings of synthetic noisy CIFAR datasets**    The generation of symmetric noisy dataset is adopted from (Cheng et al., 2020). The symmetric noise rates are $[0.2, 0.4, 0.6]$. We choose two methods to train GLS.

- **Direct training:** this setting is the same as training on clean CIFAR-10 dataset.
- **Warm-up:** in this case, we firstly train a ResNet34 model with Cross-Entropy loss for 120 epochs. For this warm-up, the only difference in hyper-parameter setting is the learning rate, where the initial learning rate is 0.1 and it multiplies 0.1 for every 40 epochs. After the warm-up, GLS loads the same pre-trained model and trains for 100 epochs with learning rate 1e-6.

### D.2    WHY NLS IS OVERLOOKED?

When learning from a relative large scale dataset, NLS tends to push the model become overly confident early in the training. The poor performances of NLS (direct-train) in Table 6 explain why NLS is neglected. When there is no warm-up, training NLS directly without warming up will reach a $88\% - 92\%$ test accuracy on the clean data. The performance will degrade much more significantly than PLS when the noise level is high or $|r|$ is large. In Table 6, we provide the comparisons between direct-train and warm-up in several settings. The improvement bring by a warm-up procedure becomes much more significantly in the high noise regime. NLS makes the classifier be overly confident at the early training which results in converging to a bad local optimum (without CE warm-up, NLS frequently results in a worse performance in CIFAR-10 and CIFAR-100). Since the model will usually fit on the clean data first, then over-fits on the noisy ones (Liu et al., 2020), a large number of approaches (such as Loss corrections (Patrini et al., 2017), Peer Loss (Liu & Guo, 2020), etc) adopt a CE warm-up firstly. Note that there is no difference in the computing costs between NLS (with CE warmup) and CE loss, proceeding with NLS to enhance the model confidence makes NLS much more competitive in the high noise regime, also gives practical insights on how to make NLS work better when learning with clean data.

Table 6: Test accuracies of GLS on assymetric noisy CIFAR-10 and symmetric CIFAR-100 (left/right denotes direct train / warm-up).

| Smooth Rate | Cifar-10 Asymmetric | | CIFAR-100 Symmetric | |
| :---: | :---: | :---: | :---: | :---: |
| | $\varepsilon = 0.2$ | $\varepsilon = 0.3$ | $\varepsilon = 0.4$ | $\varepsilon = 0.6$ |
| $r = 0.8$ | 87.89 / 90.51 | 86.38 / 87.97 | 54.78 / 51.27 | 40.21 / 39.80 |
| $r = 0.6$ | 89.14 / 90.55 | 85.97 / 88.01 | 52.83 / 52.88 | 39.64 / 40.57 |
| $r = 0.4$ | 88.23 / **90.61** | 86.95 / 88.04 | 51.40 / 54.36 | 38.29 / 41.63 |
| $r = -0.4$ | 19.71 / 89.60 | 21.86 / **88.42** | 40.30 / 56.97 | 31.35 / 43.91 |
| $r = -0.8$ | - / 89.02 | - / 88.28 | 22.63 / 57.45 | 26.75 / 44.19 |
| $r = -1.0$ | - / 88.68 | - / 88.29 | - / 57.53 | - / 44.59 |
| $r = -2.0$ | - / 88.86 | - / 88.13 | - / 58.21 | - / 45.47 |
| $r = -4.0$ | - / 89.80 | - / 88.20 | - / **58.47** | - / 46.86 |
| $r = -6.0$ | - / 90.02 | - / 88.18 | - / 57.87 | - / **47.18** |

### D.3 Experiment details on synthetic datasets and UCI

We introduce experiment details on synthetic datasets and UCI datasets adopted in our experiment designs.

**Generation of synthetic dataset** In the synthetic (Type 1) dataset, we generate 500 points for both classes. Class +1 distributes inside the circle with radius 0.25. Class -1 generates by randomly sampling 500 data points in the annulus with inner radius 0.28 and outer radius 0.45. As for synthetic (Type 2) dataset, we uniformly assign labels for 50% samples in the annulus (with inner radius 0.22, outer radius 0.31) based on Type 1 dataset.

**Generating noise labels on synthetic datasets and UCI datasets** Note that these datasets are all binary classification datasets, each label in the training and validation set is flipped to the other class with probability $e$, and we set $e = 0.1, 0.4$ for synthetic Type 1 dataset, $e = 0.1, 0.3$ for synthetic Type 2 dataset.

**Training settings of synthetic datasets** For both types of synthetic datasets, we adopted a three-layer ReLU Multi-Layer Perceptron (MLP), trained for 200 epochs with batch-size 128 and Adam (Kingma & Ba, 2014) optimizer. The initial learning rate is 0.1, and it multiplies 0.1 for every 40 epochs.

**Training settings of UCI datasets Dua & Graff (2017)** We adopted (Liu & Guo, 2020) a two-layer ReLU Multi-Layer Perceptron (MLP) for classification tasks on multiple UCI datasets, trained for 1000 episodes with batch-size 64 and Adam (Kingma & Ba, 2014) optimizer. We report the best performance for each smooth rate under a set of learning rate settings, $[0.0007, 0.001, 0.005, 0.01, 0.05]$.

### D.4 Additional experiment on $r^*$ and $r_{\text{OPT}}$

$r^*$ **and $r_{\text{opt}}$ on synthetic dataset** We generate 2D (binary) synthetic dataset by randomly sampling two circularly distributed classes. The inner annulus indicates one class (blue), while the outer annulus denotes the other class (red). Clearly, the generated synthetic dataset is well-separable (Type 1) and we hold 20% data samples for performance comparison. The noise transition matrix takes a symmetric form with noise rate $e_i$ for both classes. To simulate the scenario where the clean data may not be perfectly separated due to a non-negligible amount of uncertainty samples clustering at the decision boundary, we flip the label of 50% samples near the intersection of two annulus to the other class (Type 2). As specified in Table 7, $r^* = [0.1, 0.4]$ for Type 1 data and $r^* = [0.0, 0.2]$ for Type 2 data. With the presence of label noise, the distribution of $r_{\text{opt}}$ shifts from non-negative ones to negative values. Even though NLS fails to outperform PLS on clean data, we observe that NLS is less sensitive to noisy labels. Data with high level noise rates clearly favor NLS with a low smooth rate!

Table 7: Test accuracies of GLS on clean and noisy synthetic data. We report best test accuracy for each method. $r_{\text{opt}}$ and the corresponding test accuracy are highlighted (green: NLS; red: PLS).

| Method | Synthetic data (Type 1) | | | Synthetic data (Type 2) | | |
|---|---|---|---|---|---|---|
| | $e_i = 0$ | $e_i = 0.2$ | $e_i = 0.4$ | $e_i = 0$ | $e_i = 0.2$ | $e_i = 0.4$ |
| PLS | 0.896 | 0.878 | 0.786 | 0.894 | 0.848 | 0.842 |
| Vanilla Loss | 0.889 | 0.882 | 0.806 | 0.894 | 0.875 | 0.868 |
| NLS | 0.893 | 0.885 | 0.825 | 0.883 | 0.884 | 0.875 |
| $r_{\text{opt}} =$ | [0.1, 0.4] | -0.2 | -0.4 | [0, 0.2] | -0.3 | -0.5 |

$r^*$ **and $r_{\text{opt}}$ on more UCI datasets** We further test the performance of generalized label smoothing on 7 more UCI datasets (Heart, Breast 1, Breast 2, Diabetes, German, Image and Waveform). Our observation remains unchanged: there exists a general trend that with the increasing of noise rates, NLS becomes much more competitive than PLS. Here, we attach the results of 4 additional UCI datasets for illustration.

The noisy labels are generated by a symmetric noise transition matrix with noise rate $e_i = [0.1, 0.2, 0.3, 0.4]$. As highlighted in Table 8, $r_{\text{opt}}$ appears with positive values when the data is clean (same as $r^*$) or of a low

Table 8: Test accuracies of GLS on clean and noisy UCI datasets (Image, Waveform, Heart, Banana) with best two smooth rates (green: NLS; red: PLS).

| Smooth Rate | Image | | | | | Waveform | | | | |
|---|---|---|---|---|---|---|---|---|---|---|
| | $e_i = 0$ | $e_i = 0.1$ | $e_i = 0.2$ | $e_i = 0.3$ | $e_i = 0.4$ | $e_i = 0$ | $e_i = 0.1$ | $e_i = 0.2$ | $e_i = 0.3$ | $e_i = 0.4$ |
| $r = 0.8$ | 0.993 | 0.983 | 0.973 | 0.946 | 0.875 | 0.939 | 0.935 | 0.931 | 0.927 | 0.885 |
| $r = 0.6$ | 0.993 | 0.987 | 0.970 | 0.939 | 0.869 | 0.943 | 0.943 | 0.943 | 0.929 | 0.901 |
| $r = 0.4$ | 0.997 | 0.980 | 0.973 | 0.939 | 0.865 | 0.941 | 0.937 | 0.943 | 0.931 | 0.905 |
| $r = 0.2$ | 0.993 | 0.993 | 0.966 | 0.936 | 0.875 | 0.941 | 0.935 | 0.933 | 0.931 | 0.913 |
| $r = 0.0$ | 0.990 | 0.976 | 0.963 | 0.929 | 0.865 | 0.945 | 0.935 | 0.937 | 0.933 | 0.911 |
| $r = -0.2$ | 0.912 | 0.96 | 0.953 | 0.919 | 0.872 | 0.937 | 0.939 | 0.939 | 0.933 | 0.907 |
| $r = -0.4$ | 0.882 | 0.923 | 0.953 | 0.936 | 0.872 | 0.925 | 0.937 | 0.939 | 0.933 | 0.917 |
| $r = -0.8$ | 0.842 | 0.882 | 0.926 | 0.933 | 0.872 | 0.921 | 0.925 | 0.939 | 0.931 | 0.923 |
| $r = -1.0$ | 0.832 | 0.869 | 0.909 | 0.929 | 0.882 | 0.921 | 0.923 | 0.933 | 0.929 | 0.907 |
| $r = -2.0$ | 0.818 | 0.815 | 0.889 | 0.909 | 0.906 | 0.911 | 0.913 | 0.921 | 0.927 | 0.911 |
| Smooth Rate | Heart | | | | | Banana | | | | |
| | $e_i = 0$ | $e_i = 0.1$ | $e_i = 0.2$ | $e_i = 0.3$ | $e_i = 0.4$ | $e_i = 0$ | $e_i = 0.1$ | $e_i = 0.2$ | $e_i = 0.3$ | $e_i = 0.4$ |
| $r = 0.8$ | 0.885 | 0.853 | 0.836 | 0.820 | 0.738 | 0.896 | 0.893 | 0.876 | 0.847 | 0.790 |
| $r = 0.6$ | 0.902 | 0.836 | 0.820 | 0.836 | 0.738 | 0.903 | 0.881 | 0.876 | 0.855 | 0.811 |
| $r = 0.4$ | 0.885 | 0.853 | 0.836 | 0.820 | 0.771 | 0.900 | 0.887 | 0.874 | 0.859 | 0.807 |
| $r = 0.2$ | 0.902 | 0.853 | 0.820 | 0.803 | 0.754 | 0.896 | 0.894 | 0.876 | 0.856 | 0.810 |
| $r = 0.0$ | 0.902 | 0.853 | 0.820 | 0.820 | 0.771 | 0.897 | 0.881 | 0.871 | 0.849 | 0.833 |
| $r = -0.4$ | 0.869 | 0.836 | 0.803 | 0.853 | 0.754 | 0.847 | 0.874 | 0.859 | 0.853 | 0.840 |
| $r = -0.6$ | 0.869 | 0.836 | 0.820 | 0.853 | 0.721 | 0.845 | 0.864 | 0.861 | 0.859 | 0.837 |
| $r = -1.0$ | 0.885 | 0.869 | 0.803 | 0.853 | 0.754 | 0.796 | 0.812 | 0.852 | 0.854 | 0.811 |
| $r = -2.0$ | 0.885 | 0.869 | 0.820 | 0.853 | 0.787 | 0.759 | 0.764 | 0.819 | 0.852 | 0.819 |
| $r = -4.0$ | 0.885 | 0.869 | 0.853 | 0.885 | 0.820 | 0.718 | 0.723 | 0.738 | 0.787 | 0.813 |
| $r = -8.0$ | 0.869 | 0.869 | 0.885 | 0.853 | 0.853 | 0.703 | 0.700 | 0.699 | 0.735 | 0.735 |

noise rate. With the increasing of noise rates, NLS becomes more competitive than PLS. We color-code different noise regimes where either PLS (red-ish) or NLS (green-ish) outperforms the other. Clearly, there is a separation of the favored smoothing rate for different noise scenarios (upper left & low noise for PLS, bottom right & high noise for NLS).

## D.5 ADDITIONAL EXPERIMENT RESULTS ON MODEL CONFIDENCE

**NLS improves model confidence on Synthetic Type 2 dataset** In this case, the clean data that are close to decision boundary distributes randomly. In Figure 5-6, the colored bands depict the different levels of prediction probabilities. When the smooth rate increases from negative to positive, more samples fall in the orange and light blue band which indicates uncertain predictions. When the smooth rate increases from negative to positive, learning with GLS will result in more uncertain predictions. With the increasing of noise rates ($e_i = 0 \rightarrow 0.4$), GLS with a fixed smooth rate becomes less confident on its predictions. Thus, a smaller smooth rate is required when the noise rate increases.

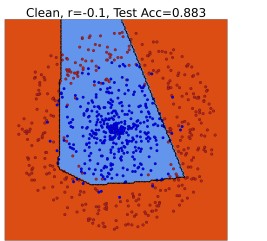 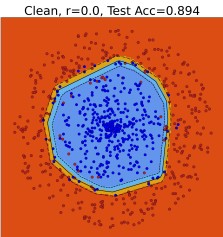 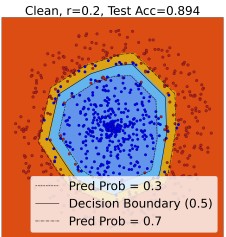

Figure 5: Model confidence visualization of GLS on synthetic data (Type 2) with the clean data. $r^* \in [0, 0.2]$. (left: NLS; middle: Vanilla Loss; right: PLS).

**Model confidence of GLS on CIFAR-10 test dataset** When GLS trained on symmetric 0.2 noisy CIFAR-10 training dataset (see Figure 7), with the decreasing of smooth rates (from right to left), the model

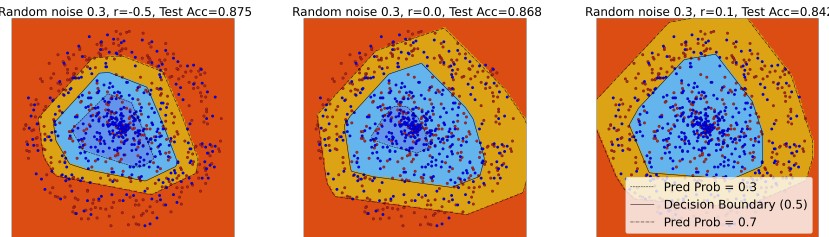

Figure 6: Model confidence visualization of GLS on synthetic data (Type 2) with noise rate $e_i = 0.3$. $r_{\mathrm{opt}} = -0.5$. (left: NLS; middle: Vanilla Loss; right: PLS).

confidence on correct predictions gradually approach to its maximum, while for wrong predictions, the model confidence converges to its minimum value. NLS makes the model prediction become over-confident on correct predictions and in-confident on wrong predictions.

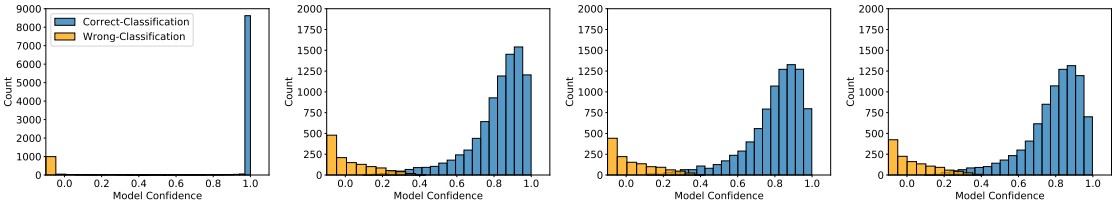

Figure 7: Model confidence distribution of correct and wrong predictions on CIFAR-10 test data. (From left to right: NLS ($r = -0.8, -0.4$), Vanilla Loss, PLS ($r = 0.4$), trained on symmetric 0.2 noisy CIFAR-10 dataset).

### D.6 EFFECT OF GLS ON PRE-LOGITS

We visualise the pre-logits of a ResNet-34 for three classes on CIFAR-10. We adopt the method from (Müller et al., 2019) which illustrates how representations differ between penultimate layers of networks trained with different smooth rates in GLS. In Figure 8, NLS makes the model $f$ be confident on her predictions and the distances between three clusters are clearly larger than those appeared in Vanilla Loss and PLS.

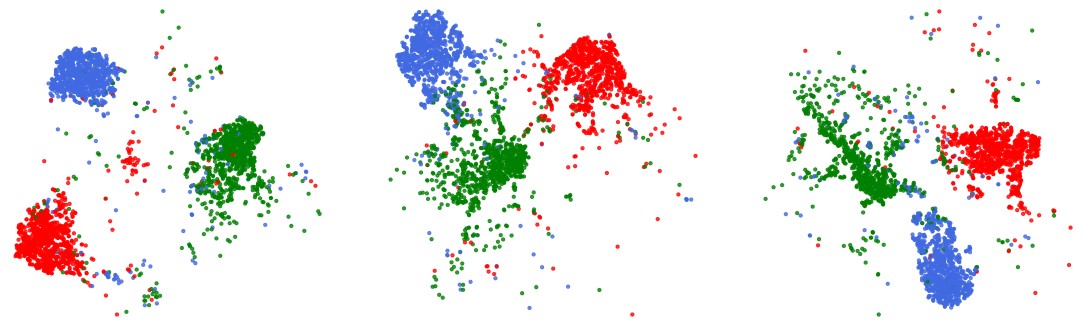

Figure 8: Effect of GLS on pre-logits (left: NLS; middle: Vanilla Loss; right: PLS; trained with symmetric 0.2 noisy CIFAR-10 training dataset).

## E   BIAS AND VARIANCE TRADE-OFF OF GLS

Denote $\hat{f}_H$, $\hat{f}_S$ as pre-trained models on the training dataset $D$ w.r.t. hard labels and soft labels, respectively. The vector form of the prediction w.r.t. sample $x$ given by $\hat{f}_H$ and $\hat{f}_S$ are $\hat{\mathbf{f}}_\mathbf{H}(x; D)$ and $\hat{\mathbf{f}}_\mathbf{S}(x; D)$. For the ease of presentation, we relate notations with subscript H/S to hard/soft labels without further explanation. Given the sample $x$ and the one-hot label $\mathbf{y}$, we denote the averaged model prediction by:

$$\overline{\mathbf{f}}_\mathbf{H}(x; D) := \frac{1}{Z_H} \exp\left[\mathbb{E}_D \log(\hat{\mathbf{f}}_\mathbf{H}(x; D))\right], \quad \overline{\mathbf{f}}_\mathbf{S}(x; D) := \frac{1}{Z_S} \exp\left[\mathbb{E}_D \log(\hat{\mathbf{f}}_\mathbf{S}(x; D))\right]$$

where $Z_H$, $Z_S$ are normalization constants. The bias of model prediction is defined as the KL divergence $D_{KL}$ between target distribution (one-hot encoded vector form) $\mathbf{y}$ and the averaged model prediction.

$$\text{Bias}_H := \mathbb{E}_{x,\mathbf{y}}\left[\mathbf{y} \log \frac{\mathbf{y}}{\overline{\mathbf{f}}_\mathbf{H}(x; D)}\right], \quad \text{Bias}_S := \mathbb{E}_{x,\mathbf{y}}\left[\mathbf{y} \log \frac{\mathbf{y}}{\overline{\mathbf{f}}_\mathbf{S}(x; D)}\right]$$

While the variance of model prediction measures the expectation of KL divergence between the averaged model prediction and model prediction over $D$:

$$\text{Var}_H := \mathbb{E}_D\left[\mathbb{E}_{x,\mathbf{y}}\left[\overline{\mathbf{f}}_\mathbf{H}(x; D) \log \left(\frac{\overline{\mathbf{f}}_\mathbf{H}(x; D)}{\hat{\mathbf{f}}_\mathbf{H}(x; D)}\right)\right]\right], \quad \text{Var}_S := \mathbb{E}_D\left[\mathbb{E}_{x,\mathbf{y}}\left[\overline{\mathbf{f}}_\mathbf{S}(x; D) \log \left(\frac{\overline{\mathbf{f}}_\mathbf{S}(x; D)}{\hat{\mathbf{f}}_\mathbf{S}(x; D)}\right)\right]\right]$$

Empirical observation from (Zhou et al., 2021) shows that the variance brought by learning with positive soft labels given by a teacher's model (Hinton et al., 2015) is less than the direct training w.r.t hard labels. As an extension, we are interested in how GLS interferes with the bias and variance of model prediction.

**Bias and variance of GLS on clean dataset**   We introduce our empirical observation regarding the role of GLS in bias and variance trade-off in Figure 9. We select nine smooth rates of GLS for illustration. Each smooth rate setting of GLS trains on the CIFAR-10 dataset for 5 times with different data augmentations. To estimate the variance and bias of pre-trained models, we adopt the implementation in (Yang et al., 2020). Empirical results show that learning directly with a larger positive smooth rate typically results in lower variance and higher bias. In Figure 9, we can observe almost constant bias values and very low variance for NLS. This is best explained by the warm-up of pre-trained models and the fact that NLS pushes the classifier to give confident predictions. As for PLS, with the increase of smooth rate, the overall bias has an increasing tendency while the variance has the decreasing pattern. Especially when the smooth rate approaches to 1, i.e., $r = 0.9$, the variance is close to 0.

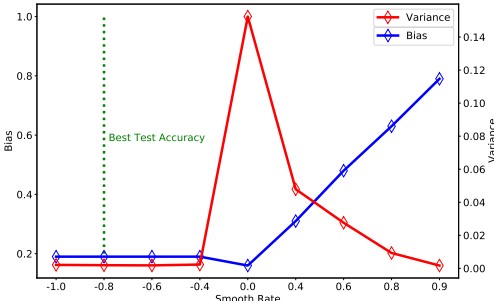

Figure 9: Bias and variance of pre-trained GLS models on clean CIFAR-10 test dataset.

# F OMITTED PROOFS

## F.1 PROOF OF THEOREM 1

Before we prove Theorem 1, we first introduce Lemma 1.

**Lemma 1.** $\forall (x, \mathbf{y}^{GLS,r})$, $\ell\big(\mathbf{f}(\mathbf{x}), \mathbf{y}^{GLS,r}\big) = \big(1 - \frac{r}{2}\big) \cdot \ell\big(\mathbf{f}(\mathbf{x}), y\big) + \frac{r}{2} \cdot \ell\big(\mathbf{f}(\mathbf{x}), 1 - y\big)$.

**Proof of Lemma 1**

*Proof.* For CE loss, due to its linear property w.r.t. the label, we directly have:

$$\ell(\mathbf{f}(\mathbf{x}), \mathbf{y}^{\text{GLS},r}) = \ell\big(\mathbf{f}(\mathbf{x}), \mathbf{y} \cdot (1 - r) + \frac{r}{2}\big) = \big(1 - \frac{r}{2}\big) \cdot \ell\big(\mathbf{f}(\mathbf{x}), y\big) + \frac{r}{2} \cdot \ell\big(\mathbf{f}(\mathbf{x}), 1 - y\big)$$

$\square$

**Proof of Theorem 1**

*Proof.* Based on Lemma 1, with a bit of math, for NLS, we have:

$$
\begin{aligned}
&\min \mathbb{E}_{(X,\widetilde{Y}) \sim \widetilde{\mathcal{D}}} \Big[ \ell\big(\mathbf{f}(\mathbf{X}), \widetilde{Y}^{\text{GLS},-r}\big) \Big] \\
=& \min \mathbb{E}_{(X,\widetilde{Y}) \sim \widetilde{\mathcal{D}}} \Big[ \big(1 + \frac{r}{2}\big) \cdot \ell\big(\mathbf{f}(\mathbf{X}), \widetilde{Y}\big) - \frac{r}{2} \cdot \ell\big(\mathbf{f}(\mathbf{X}), 1 - \widetilde{Y}\big) \Big] \\
=& \min \mathbb{E}_{(X,\widetilde{Y}) \sim \widetilde{\mathcal{D}}} \Big[ \big[\big(1 + \frac{r}{2}\big) + \big(1 - \frac{r}{2}\big)\big] \cdot \ell\big(\mathbf{f}(\mathbf{X}), \widetilde{Y}\big) - \big[\big(1 - \frac{r}{2}\big) \cdot \ell\big(\mathbf{f}(\mathbf{X}), \widetilde{Y}\big) + \frac{r}{2} \cdot \ell\big(\mathbf{f}(\mathbf{X}), 1 - \widetilde{Y}\big)\big] \Big] \\
=& \min \mathbb{E}_{(X,\widetilde{Y}) \sim \widetilde{\mathcal{D}}} \Big[ 2 \cdot \ell\big(\mathbf{f}(\mathbf{X}), \widetilde{Y}\big) - \ell\big(\mathbf{f}(\mathbf{X}), \widetilde{Y}^{\text{GLS},r}\big) \Big]
\end{aligned}
$$

$\square$

## F.2 Proof of Theorem 5

*Proof.*

$$
\begin{aligned}
Eqn.3 &= \min \mathbb{E}_{(X,\widetilde{Y})\sim\widetilde{\mathcal{D}}}\Big[\underbrace{\big(1-\tfrac{r}{2}\big)}_{:=c_1}\cdot\ell\big(\mathbf{f}(\mathbf{X}),\widetilde{Y}\big) + \underbrace{\tfrac{r}{2}}_{:=c_2}\cdot\ell\big(\mathbf{f}(\mathbf{X}),1-\widetilde{Y}\big)\Big] \\
&= \min \mathbb{E}_{X,Y=0}\Big[\mathbb{P}(\widetilde{Y}=0|Y=0)\cdot\big(c_1\cdot\ell\big(\mathbf{f}(\mathbf{X}),0\big)+c_2\cdot\ell\big(\mathbf{f}(\mathbf{X}),1\big)\big) \\
&\quad + \mathbb{P}(\widetilde{Y}=1|Y=0)\cdot\big(c_1\cdot\ell\big(\mathbf{f}(\mathbf{X}),1\big)+c_2\cdot\ell\big(\mathbf{f}(\mathbf{X}),0\big)\big)\Big] \\
&\quad + \mathbb{E}_{X,Y=1}\Big[\mathbb{P}(\widetilde{Y}=0|Y=1)\cdot\big(c_1\cdot\ell\big(\mathbf{f}(\mathbf{X}),0\big)+c_2\cdot\ell\big(\mathbf{f}(\mathbf{X}),1\big)\big) \\
&\quad + \mathbb{P}(\widetilde{Y}=1|Y=1)\cdot\big(c_1\cdot\ell\big(\mathbf{f}(\mathbf{X}),1\big)+c_2\cdot\ell\big(\mathbf{f}(\mathbf{X}),0\big)\big)\Big] \\
&= \min \mathbb{E}_{X,Y=0}\Big[\big[(1-e_0)\cdot c_1+e_0\cdot c_2\big]\cdot\ell\big(\mathbf{f}(\mathbf{X}),0\big)+\big[(1-e_0)\cdot c_2+e_0\cdot c_1\big]\cdot\ell\big(\mathbf{f}(\mathbf{X}),1\big)\Big] \\
&\quad + \mathbb{E}_{X,Y=1}\Big[\big[(1-e_1)\cdot c_1+e_1\cdot c_2\big]\cdot\ell\big(\mathbf{f}(\mathbf{X}),1\big)+\big[(1-e_1)\cdot c_2+e_1\cdot c_1\big]\cdot\ell\big(\mathbf{f}(\mathbf{X}),0\big)\Big] \\
&= \min \mathbb{E}_{X,Y=0}\Big[\big[(1-e_0)\cdot c_1+e_0\cdot c_2\big]\cdot\ell\big(\mathbf{f}(\mathbf{X}),0\big)+\big[(1-e_0)\cdot c_2+e_0\cdot c_1\big]\cdot\ell\big(\mathbf{f}(\mathbf{X}),1\big)\Big] \\
&\quad + \mathbb{E}_{X,Y=1}\Big[\big[(1-e_0)\cdot c_1+e_0\cdot c_2\big]\cdot\ell\big(\mathbf{f}(\mathbf{X}),1\big)+\big[(1-e_0)\cdot c_2+e_0\cdot c_1\big]\cdot\ell\big(\mathbf{f}(\mathbf{X}),0\big)\Big] \\
&\quad + \mathbb{E}_{X,Y=1}\Big[e_\Delta\cdot(c_2-c_1)\cdot\ell\big(\mathbf{f}(\mathbf{X}),1\big)-e_\Delta\cdot(c_2-c_1)\cdot\ell\big(\mathbf{f}(\mathbf{X}),0\big)\Big] \\
&= \min \mathbb{E}_{(X,Y)\sim\mathcal{D}}\Big[\big[(1-e_0)\cdot c_1+e_0\cdot c_2\big]\cdot\ell\big(\mathbf{f}(\mathbf{X}),Y\big)+\big[(1-e_0)\cdot c_2+e_0\cdot c_1\big]\cdot\ell\big(\mathbf{f}(\mathbf{X}),1-Y\big)\Big] \\
&\quad - e_\Delta\cdot(c_1-c_2)\cdot\mathbb{E}_{X,Y=1}\Big[\ell\big(\mathbf{f}(\mathbf{X}),1\big)-\ell\big(\mathbf{f}(\mathbf{X}),0\big)\Big] \\
&= \min \mathbb{E}_{(X,Y)\sim\mathcal{D}}\Big[(c_1+c_2)\cdot\ell\big(\mathbf{f}(\mathbf{X}),Y\big)\Big] \\
&\quad + \big[(1-e_0)\cdot c_2+e_0\cdot c_1\big]\cdot\mathbb{E}_{(X,Y)\sim\mathcal{D}}\Big[\ell\big(\mathbf{f}(\mathbf{X}),1-Y\big)-\ell\big(\mathbf{f}(\mathbf{X}),Y\big)\Big] \\
&\quad - e_\Delta\cdot(c_1-c_2)\cdot\mathbb{E}_{X,Y=1}\Big[\ell\big(\mathbf{f}(\mathbf{X}),1\big)-\ell\big(\mathbf{f}(\mathbf{X}),0\big)\Big] \\
&= \min \mathbb{E}_{(X,Y)\sim\mathcal{D}}\Big[(c_1+c_2)\cdot\ell\big(\mathbf{f}(\mathbf{X}),Y^*\big)\Big] \\
&\quad + \Big[-\tfrac{r^*}{2}+(1-e_0)\cdot c_2+e_0\cdot c_1\Big]\cdot\mathbb{E}_{(X,Y)\sim\mathcal{D}}\Big[\ell\big(\mathbf{f}(\mathbf{X}),1-Y\big)-\ell\big(\mathbf{f}(\mathbf{X}),Y\big)\Big] \\
&\quad - e_\Delta\cdot(c_1-c_2)\cdot\mathbb{E}_{X,Y=1}\Big[\ell\big(\mathbf{f}(\mathbf{X}),1\big)-\ell\big(\mathbf{f}(\mathbf{X}),0\big)\Big] \\
&= \min \underbrace{\mathbb{E}_{(X,Y)\sim\mathcal{D}}\Big[\ell\big(\mathbf{f}(\mathbf{X}),Y^*\big)\Big]}_{\text{True Risk}} + \underbrace{\lambda_1\cdot\mathbb{E}_{(X,Y)\sim\mathcal{D}}\Big[\ell\big(\mathbf{f}(\mathbf{X}),1-Y\big)-\ell\big(\mathbf{f}(\mathbf{X}),Y\big)\Big]}_{\text{M-Inc1}} \\
&\quad + \underbrace{\lambda_2\cdot\mathbb{E}_{X,Y=1}\Big[\ell\big(\mathbf{f}(\mathbf{X}),0\big)-\ell\big(\mathbf{f}(\mathbf{X}),1\big)\Big]}_{\text{M-Inc2}}
\end{aligned}
$$

$\square$

## F.3  PROOF OF PROPOSITION 1

*Proof.* The risk minimization of backward correction is equivalent to:

$$\mathbb{E}_{(X,\widetilde{Y})\sim\widetilde{\mathcal{D}}}\Big[\ell^{\leftarrow}\big(\mathbf{f}(\mathbf{X}),\widetilde{Y}\big)\Big] =\mathbb{E}_{(X,Y)\sim\mathcal{D}}\Big[\ell\big(\mathbf{f}(\mathbf{X}),Y\big)\Big] \quad \text{(By Theorem 1 in (Patrini et al., 2017))}$$

The risk minimization of forward correction is equivalent to:

$$\mathbb{E}_{(X,\widetilde{Y})\sim\widetilde{\mathcal{D}}}\Big[\ell^{\rightarrow}\big(\mathbf{f}(\mathbf{X}),\widetilde{Y}\big)\Big] =\mathbb{E}_{(X,Y)\sim\mathcal{D}}\Big[\ell\big(\mathbf{f}(\mathbf{X}),Y\big)\Big] \quad \text{(By Theorem 2 in (Patrini et al., 2017))}$$

Theorem 1 and 2 in (Patrini et al., 2017) demonstrate that forward and backward corrected losses equal the original loss $\ell$ computed on the clean data in expectation. Thus, for $r_{\text{LC}} = \frac{2e_0}{2e_0-1}$, by Theorem 5 (adopt $r^* = 0$), we have:

$$\min \mathbb{E}_{(X,\widetilde{Y})\sim\widetilde{\mathcal{D}}}\quad\Big[\ell\big(\mathbf{f}(\mathbf{X}),\widetilde{Y}^{\text{GLS},r_{\text{LC}}}\big)\Big] + \lambda_{\text{LC}} \cdot \underbrace{\mathbb{E}_{X,Y=1}\Big[\ell\big(\mathbf{f}(\mathbf{X}),1\big) - \ell\big(\mathbf{f}(\mathbf{X}),0\big)\Big]}_{\text{Bias-LC}}$$

$$= \min \mathbb{E}_{(X,Y)\sim\mathcal{D}}\Big[\ell\big(\mathbf{f}(\mathbf{X}),Y\big)\Big] + \Big[e_0 + (1-2e_0)\cdot\frac{r_{\text{LC}}}{2}\Big]\cdot\mathbb{E}_{(X,Y)\sim\mathcal{D}}\Big[\ell\big(\mathbf{f}(\mathbf{X}),1-Y\big) - \ell\big(\mathbf{f}(\mathbf{X}),Y\big)\Big]$$

$$+ e_\Delta \cdot (1-r_{\text{LC}}) \cdot \mathbb{E}_{X,Y=1}\Big[\ell\big(\mathbf{f}(\mathbf{X}),0\big) - \ell\big(\mathbf{f}(\mathbf{X}),1\big)\Big] + \lambda_{\text{LC}} \cdot \mathbb{E}_{X,Y=1}\Big[\ell\big(\mathbf{f}(\mathbf{X}),1\big) - \ell\big(\mathbf{f}(\mathbf{X}),0\big)\Big]$$

$$= \min \mathbb{E}_{(X,Y)\sim\mathcal{D}}\Big[\ell\big(\mathbf{f}(\mathbf{X}),Y\big)\Big] + e_\Delta \cdot \Big(\frac{1}{1-2e_0} - \frac{1}{1-2e_0}\Big)\cdot\mathbb{E}_{X,Y=1}\Big[\ell\big(\mathbf{f}(\mathbf{X}),0\big) - \ell\big(\mathbf{f}(\mathbf{X}),1\big)\Big]$$

$$= \min \mathbb{E}_{(X,Y)\sim\mathcal{D}}\Big[\ell\big(\mathbf{f}(\mathbf{X}),Y\big)\Big]$$

Thus,

$$\min \mathbb{E}_{(X,\widetilde{Y})\sim\widetilde{\mathcal{D}}}\Big[\ell^{\leftarrow}\big(\mathbf{f}(\mathbf{X}),\widetilde{Y}\big)\Big] = \min \mathbb{E}_{(X,\widetilde{Y})\sim\widetilde{\mathcal{D}}}\Big[\ell^{\rightarrow}\big(\mathbf{f}(\mathbf{X}),\widetilde{Y}\big)\Big]$$

$$= \min \mathbb{E}_{(X,\widetilde{Y})\sim\widetilde{\mathcal{D}}}\Big[\ell\big(\mathbf{f}(\mathbf{X}),\widetilde{Y}^{\text{GLS},r_{\text{LC}}}\big)\Big] + \lambda_{\text{LC}} \cdot \underbrace{\mathbb{E}_{X,Y=1}\Big[\ell\big(\mathbf{f}(\mathbf{X}),1\big) - \ell\big(\mathbf{f}(\mathbf{X}),0\big)\Big]}_{\text{Bias-LC}}$$

$$\square$$

## F.4  PROOF OF THEOREM 2

*Proof.* Based on Proposition 1, when $e_\Delta = 0$, $\lambda_{\text{LC}} = 0$, we directly have:

$$\min \mathbb{E}_{(X,\widetilde{Y})\sim\widetilde{\mathcal{D}}}\Big[\ell^{\leftarrow}\big(\mathbf{f}(\mathbf{X}),\widetilde{Y}\big)\Big] = \min \mathbb{E}_{(X,\widetilde{Y})\sim\widetilde{\mathcal{D}}}\Big[\ell^{\rightarrow}\big(\mathbf{f}(\mathbf{X}),\widetilde{Y}\big)\Big]$$

$$= \min \mathbb{E}_{(X,\widetilde{Y})\sim\widetilde{\mathcal{D}}}\Big[\ell\big(\mathbf{f}(\mathbf{X}),\widetilde{Y}^{\text{GLS},r_{\text{LC}}}\big)\Big]$$

$$\square$$

## F.5  PROOF OF THEOREM 3

*Proof.* Note that

$$\min \mathbb{E}_{(X,\widetilde{Y})\sim\widetilde{\mathcal{D}}}\Big[\ell_{\text{CL}}\big(\mathbf{f}(\mathbf{X}),\widetilde{Y}\big)\Big] = \min \mathbb{E}_{(X,\widetilde{Y})\sim\widetilde{\mathcal{D}}}\Big[\ell\big(\mathbf{f}(\mathbf{X}),\widetilde{Y}\big) - \ell\big(\mathbf{f}(\mathbf{X}),1-\widetilde{Y}\big)\Big]$$

We have:

$$\min \mathbb{E}_{(X,\widetilde{Y})\sim\widetilde{\mathcal{D}}}\Big[\ell\big(\mathbf{f}(\mathbf{X}),\widetilde{Y}^{\text{GLS},r_{\text{CL}}}\big)\Big]$$
$$= \min \mathbb{E}_{(X,\widetilde{Y})\sim\widetilde{\mathcal{D}}}\Big[\big(1-\frac{r_{\text{CL}}}{2}\big)\cdot\ell\big(\mathbf{f}(\mathbf{X}),\widetilde{Y}\big)+\frac{r_{\text{CL}}}{2}\cdot\ell\big(\mathbf{f}(\mathbf{X}),1-\widetilde{Y}\big)\Big]$$
$$\Longleftrightarrow \min \mathbb{E}_{(X,\widetilde{Y})\sim\widetilde{\mathcal{D}}}\Big[\ell\big(\mathbf{f}(\mathbf{X}),\widetilde{Y}\big)+\frac{r_{\text{CL}}}{2-r_{\text{CL}}}\cdot\ell\big(\mathbf{f}(\mathbf{X}),1-\widetilde{Y}\big)\Big]$$

When $r_{\text{CL}}\to-\infty$, we have $\frac{r_{\text{CL}}}{2-r_{\text{CL}}}\to-1$. Thus,

$$\min \mathbb{E}_{(X,\widetilde{Y})\sim\widetilde{\mathcal{D}}}\Big[\ell_{\text{CL}}\big(\mathbf{f}(\mathbf{X}),\widetilde{Y}\big)\Big] = \min \mathbb{E}_{(X,\widetilde{Y})\sim\widetilde{\mathcal{D}}}\Big[\ell\big(\mathbf{f}(\mathbf{X}),\widetilde{Y}^{\text{GLS},r_{\text{CL}}\to-\infty}\big)\Big]$$

$\square$

### F.6 PROOF OF PROPOSITION 2

*Proof.* Note that:

$$\mathbb{E}_{(X,\widetilde{Y})\sim\widetilde{\mathcal{D}}}\Big[\ell\big(\mathbf{f}(\mathbf{X}),\widetilde{Y}\big)\Big]-\mathbb{E}_{(X,\widetilde{Y})\sim\widetilde{\mathcal{D}}}\Big[\ell\big(\mathbf{f}(\mathbf{X}),\widetilde{Y}^{\text{GLS},r}\big)\Big]$$
$$=\mathbb{E}_{(X,\widetilde{Y})\sim\widetilde{\mathcal{D}}}\Big[1-\big(1-\frac{r}{2}\big)\cdot\ell\big(\mathbf{f}(\mathbf{X}),\widetilde{Y}\big)-\frac{r}{2}\cdot\ell\big(\mathbf{f}(\mathbf{X}),1-\widetilde{Y}\big)\Big]$$
$$=\frac{r}{2}\cdot\mathbb{E}_{(X,\widetilde{Y})\sim\widetilde{\mathcal{D}}}\Big[\ell\big(\mathbf{f}(\mathbf{X}),\widetilde{Y}\big)-\ell\big(\mathbf{f}(\mathbf{X}),1-\widetilde{Y}\big)\Big]$$

And we have:

$$\mathbb{E}_{(X_i,\widetilde{Y}_i)\sim\widetilde{\mathcal{D}}}\Big[\ell\big(\mathbf{f}(\mathbf{X_1}),\tilde{Y_2}\big)\Big]$$
$$=\mathbb{E}_X\Big[\mathbb{P}(\widetilde{Y}=0)\cdot\ell\big(\mathbf{f}(\mathbf{X}),0\big)+\big(1-\mathbb{P}(\widetilde{Y}=0)\big)\cdot\ell\big(\mathbf{f}(\mathbf{X}),1\big)\Big]$$
$$=\mathbb{E}_{X,\widetilde{Y}=0}\Big[\mathbb{P}(\widetilde{Y}=0)\cdot\ell\big(\mathbf{f}(\mathbf{X}),0\big)+\big(1-\mathbb{P}(\widetilde{Y}=0)\big)\cdot\ell\big(\mathbf{f}(\mathbf{X}),1\big)\Big]$$
$$\quad+\mathbb{E}_{X,\widetilde{Y}=1}\Big[\mathbb{P}(\widetilde{Y}=0)\cdot\ell\big(\mathbf{f}(\mathbf{X}),0\big)+\big(1-\mathbb{P}(\widetilde{Y}=0)\big)\cdot\ell\big(\mathbf{f}(\mathbf{X}),1\big)\Big]$$
$$=\mathbb{E}_{X,\widetilde{Y}=0}\Big[\mathbb{P}(\widetilde{Y}=0)\cdot\ell\big(\mathbf{f}(\mathbf{X}),0\big)+\big(1-\mathbb{P}(\widetilde{Y}=0)\big)\cdot\ell\big(\mathbf{f}(\mathbf{X}),1\big)\Big]$$
$$\quad+\mathbb{E}_{X,\widetilde{Y}=1}\Big[\big(1-\mathbb{P}(\widetilde{Y}=0)\big)\cdot\ell\big(\mathbf{f}(\mathbf{X}),0\big)+\mathbb{P}(\widetilde{Y}=0)\cdot\ell\big(\mathbf{f}(\mathbf{X}),1\big)\Big]$$
$$\quad+\big(1-2\cdot\mathbb{P}(\widetilde{Y}=0)\big)\cdot\mathbb{E}_{X,\widetilde{Y}=1}\Big[\ell\big(\mathbf{f}(\mathbf{X}),1\big)-\ell\big(\mathbf{f}(\mathbf{X}),0\big)\Big]$$

Thus,

$$
\min_f \mathbb{E}_{(X,\widetilde{Y})\sim\widetilde{\mathcal{D}}}\Big[\ell_{\mathrm{PL}}\big(\mathbf{f}(\mathbf{X}),\widetilde{Y}\big)\Big] = \min_f \mathbb{E}_{(X,\widetilde{Y})\sim\widetilde{\mathcal{D}}}\Big[\ell\big(\mathbf{f}(\mathbf{X}),\widetilde{Y}\big) - \ell\big(\mathbf{f}(\mathbf{X_1}),\tilde{Y}_2\big)\Big]
$$

$$
= \min_f \mathbb{E}_{(X,\widetilde{Y})\sim\widetilde{\mathcal{D}}}\Big[\ell\big(\mathbf{f}(\mathbf{X}),\widetilde{Y}\big)\Big] - \mathbb{E}_{(X_i,\widetilde{Y}_i)\sim\widetilde{\mathcal{D}}}\Big[\ell\big(\mathbf{f}(\mathbf{X_1}),\tilde{Y}_2\big)\Big]
$$

$$
= \min_f \mathbb{E}_{X,\widetilde{Y}=0}\Big[\ell\big(\mathbf{f}(\mathbf{X}),0\big)\Big] + \mathbb{E}_{X,\widetilde{Y}=1}\Big[\ell\big(\mathbf{f}(\mathbf{X}),1\big)\Big]
$$

$$
- \mathbb{E}_{X,\widetilde{Y}=0}\Big[\mathbb{P}(\widetilde{Y}=0)\cdot\ell\big(\mathbf{f}(\mathbf{X}),0\big) + \big(1-\mathbb{P}(\widetilde{Y}=0)\big)\cdot\ell\big(\mathbf{f}(\mathbf{X}),1\big)\Big]
$$

$$
- \mathbb{E}_{X,\widetilde{Y}=1}\Big[\big(1-\mathbb{P}(\widetilde{Y}=0)\big)\cdot\ell\big(\mathbf{f}(\mathbf{X}),0\big) + \mathbb{P}(\widetilde{Y}=0)\cdot\ell\big(\mathbf{f}(\mathbf{X}),1\big)\Big]
$$

$$
- \big(1-2\cdot\mathbb{P}(\widetilde{Y}=0)\big)\cdot\mathbb{E}_{X,\widetilde{Y}=1}\Big[\ell\big(\mathbf{f}(\mathbf{X}),1\big) - \ell\big(\mathbf{f}(\mathbf{X}),0\big)\Big]
$$

$$
= \min_f \mathbb{E}_{X,\widetilde{Y}=0}\Big[\big(1-\mathbb{P}(\widetilde{Y}=0)\big)\cdot\big[\ell\big(\mathbf{f}(\mathbf{X}),0\big) - \ell\big(\mathbf{f}(\mathbf{X}),1\big)\big]\Big]
$$

$$
+ \mathbb{E}_{X,\widetilde{Y}=1}\Big[\big(1-\mathbb{P}(\widetilde{Y}=0)\big)\cdot\big[\ell\big(\mathbf{f}(\mathbf{X}),1\big) - \ell\big(\mathbf{f}(\mathbf{X}),0\big)\big]\Big]
$$

$$
- \big(1-2\cdot\mathbb{P}(\widetilde{Y}=0)\big)\cdot\mathbb{E}_{X,\widetilde{Y}=1}\Big[\ell\big(\mathbf{f}(\mathbf{X}),1\big) - \ell\big(\mathbf{f}(\mathbf{X}),0\big)\Big]
$$

$$
= \min_f \mathbb{E}_{(X,\widetilde{Y})\sim\widetilde{\mathcal{D}}}\Big[\big(1-\mathbb{P}(\widetilde{Y}=0)\big)\cdot\big[\ell\big(\mathbf{f}(\mathbf{X}),\widetilde{Y}\big) - \ell\big(\mathbf{f}(\mathbf{X}),1-\widetilde{Y}\big)\big]\Big]
$$

$$
- \big(1-2\cdot\mathbb{P}(\widetilde{Y}=0)\big)\cdot\mathbb{E}_{X,\widetilde{Y}=1}\Big[\ell\big(\mathbf{f}(\mathbf{X}),1\big) - \ell\big(\mathbf{f}(\mathbf{X}),0\big)\Big]
$$

Thus, for $r_{\mathrm{PL}} = 2\cdot\mathbb{P}(\widetilde{Y}=1)$, $\lambda_{\mathrm{PL}} = 1 - r_{\mathrm{PL}}$, we have:

$$
\mathbb{E}_{(X,\widetilde{Y})\sim\widetilde{\mathcal{D}}}\Big[\ell_{\mathrm{PL}}(\mathbf{f}(\mathbf{X}),\widetilde{Y})\Big] - \Big[\mathbb{E}_{(X,\widetilde{Y})\sim\widetilde{\mathcal{D}}}\Big[\ell\big(\mathbf{f}(\mathbf{X}),\widetilde{Y}\big)\Big] - \mathbb{E}_{(X,\widetilde{Y})\sim\widetilde{\mathcal{D}}}\Big[\ell\big(\mathbf{f}(\mathbf{X}),\widetilde{Y}^{\mathrm{GLS},r_{\mathrm{PL}}}\big)\Big]\Big]
$$

$$
= \mathbb{E}_{(X,\widetilde{Y})\sim\widetilde{\mathcal{D}}}\Big[\big(1-\mathbb{P}(\widetilde{Y}=0)\big)\cdot\big[\ell\big(\mathbf{f}(\mathbf{X}),\widetilde{Y}\big) - \ell\big(\mathbf{f}(\mathbf{X}),1-\widetilde{Y}\big)\big]\Big]
$$

$$
- \big(1-2\cdot\mathbb{P}(\widetilde{Y}=0)\big)\cdot\mathbb{E}_{X,\widetilde{Y}=1}\Big[\ell\big(\mathbf{f}(\mathbf{X}),1\big) - \ell\big(\mathbf{f}(\mathbf{X}),0\big)\Big]
$$

$$
- \frac{r_{\mathrm{PL}}}{2}\cdot\mathbb{E}_{(X,\widetilde{Y})\sim\widetilde{\mathcal{D}}}\Big[\ell\big(\mathbf{f}(\mathbf{X}),\widetilde{Y}\big) - \ell\big(\mathbf{f}(\mathbf{X}),1-\widetilde{Y}\big)\Big]
$$

$$
= \mathbb{E}_{(X,\widetilde{Y})\sim\widetilde{\mathcal{D}}}\Big[\big(1-\mathbb{P}(\widetilde{Y}=0)-\mathbb{P}(\widetilde{Y}=1)\big)\cdot\big[\ell\big(\mathbf{f}(\mathbf{X}),\widetilde{Y}\big) - \ell\big(\mathbf{f}(\mathbf{X}),1-\widetilde{Y}\big)\big]\Big]
$$

$$
- \big(2\cdot\mathbb{P}(\widetilde{Y}=1)-1\big)\cdot\mathbb{E}_{X,\widetilde{Y}=1}\Big[\ell\big(\mathbf{f}(\mathbf{X}),1\big) - \ell\big(\mathbf{f}(\mathbf{X}),0\big)\Big]
$$

$$
= \lambda_{\mathrm{PL}}\cdot\mathbb{E}_{X,\widetilde{Y}=1}\Big[\ell\big(\mathbf{f}(\mathbf{X}),1\big) - \ell\big(\mathbf{f}(\mathbf{X}),0\big)\Big]
$$

And we can conclude that:

$$
\min \mathbb{E}_{(X,\widetilde{Y})\sim\widetilde{\mathcal{D}}}\Big[\ell_{\mathrm{PL}}(\mathbf{f}(\mathbf{X}),\widetilde{Y})\Big] = \min \mathbb{E}_{(X,\widetilde{Y})\sim\widetilde{\mathcal{D}}}\Big[\ell\big(\mathbf{f}(\mathbf{X}),\widetilde{Y}\big) - \ell\big(\mathbf{f}(\mathbf{X}),\widetilde{Y}^{\mathrm{GLS},r_{\mathrm{PL}}}\big)\Big]
$$

$$
+ \lambda_{PL}\cdot\underbrace{\mathbb{E}_{X,\widetilde{Y}=1}\Big[\ell(\mathbf{f}(\mathbf{X}),1) - \ell(\mathbf{f}(\mathbf{X}),0)\Big]}_{\text{Bias-PL}}
$$

$\square$

### F.7 PROOF OF THEOREM 4

*Proof.* When $\mathbb{P}(\widetilde{Y} = 0) = \mathbb{P}(\widetilde{Y} = 1)$, according to Proposition 2, we have $\lambda_{PL} = 0$ and:

$$
\begin{aligned}
\min \mathbb{E}_{(X,\widetilde{Y})\sim\widetilde{\mathcal{D}}}\Big[\ell_{\text{PL}}\big(\mathbf{f}(\mathbf{X}),\widetilde{Y}\big)\Big] &= \min \mathbb{E}_{(X,\widetilde{Y})\sim\widetilde{\mathcal{D}}}\Big[\ell\big(\mathbf{f}(\mathbf{X}),\widetilde{Y}\big) - \ell\big(\mathbf{f}(\mathbf{X}),\widetilde{Y}^{\text{GLS},r_{\text{PL}}}\big)\Big] \\
&= \min \mathbb{E}_{(X,\widetilde{Y})\sim\widetilde{\mathcal{D}}}\Big[\frac{r_{\text{PL}}}{2}\cdot\ell\big(\mathbf{f}(\mathbf{X}),\widetilde{Y}\big) - \frac{r_{\text{PL}}}{2}\cdot\ell\big(\mathbf{f}(\mathbf{X}),1-\widetilde{Y}\big)\Big] \\
&\iff \min \mathbb{E}_{(X,\widetilde{Y})\sim\widetilde{\mathcal{D}}}\Big[\ell\big(\mathbf{f}(\mathbf{X}),\widetilde{Y}\big) - \ell\big(\mathbf{f}(\mathbf{X}),1-\widetilde{Y}\big)\Big]
\end{aligned}
$$

When $r_{\text{PL}} \to -\infty$, we further have:

$$
\begin{aligned}
\min \mathbb{E}_{(X,\widetilde{Y})\sim\widetilde{\mathcal{D}}}\Big[\ell\big(\mathbf{f}(\mathbf{X}),\widetilde{Y}^{\text{GLS},r_{\text{PL}}}\big)\Big] &\iff \min \mathbb{E}_{(X,\widetilde{Y})\sim\widetilde{\mathcal{D}}}\Big[\ell\big(\mathbf{f}(\mathbf{X}),\widetilde{Y}\big) + \frac{r_{\text{CL}}}{2-r_{\text{CL}}}\cdot\ell\big(\mathbf{f}(\mathbf{X}),1-\widetilde{Y}\big)\Big] \\
&\iff \min \mathbb{E}_{(X,\widetilde{Y})\sim\widetilde{\mathcal{D}}}\Big[\ell\big(\mathbf{f}(\mathbf{X}),\widetilde{Y}\big) - \ell\big(\mathbf{f}(\mathbf{X}),1-\widetilde{Y}\big)\Big]
\end{aligned}
$$

Thus, Theorem 4 is proved. $\qquad\square$

### F.8 PROOF OF THEOREM 6

*Proof.* Note that the optimal $r$ that will cancel the impact of Term M-Inc1 is:

$$
r_{\text{opt}} := \frac{r^* - 2e}{1 - 2e}
$$

- When $e < \frac{r^*}{2}$, $r_{\text{opt}} > 0$. In this case, learning PLS with smooth rate $r_{\text{opt}}$ results in:

$$
\min \mathbb{E}_{(X,\widetilde{Y})\sim\widetilde{\mathcal{D}}}\Big[\ell(\mathbf{f}(\mathbf{X}),\widetilde{Y}^{\text{GLS},r=r_{\text{opt}}})\Big] = \min \mathbb{E}_{(X,Y)\sim\mathcal{D}}\Big[\ell\big(\mathbf{f}(\mathbf{X}),Y^*\big)\Big]
$$

  which yields $f_{\mathcal{D}}^*$;

- When $e = \frac{r^*}{2}$, $r_{\text{opt}} = 0$. Learning with the Vanilla Loss yields $f_{\mathcal{D}}^*$ since:

$$
\min \mathbb{E}_{(X,\widetilde{Y})\sim\widetilde{\mathcal{D}}}\Big[\ell(\mathbf{f}(\mathbf{X}),\widetilde{Y})\Big] = \min \mathbb{E}_{(X,Y)\sim\mathcal{D}}\Big[\ell\big(\mathbf{f}(\mathbf{X}),Y^*\big)\Big]
$$

- Similarly, when $e > \frac{r^*}{2}$, learning NLS with $r = r_{\text{opt}} < 0$ yields $f_{\mathcal{D}}^*$.

$\qquad\square$

### F.9 PROOF OF THEOREM 7

*Proof.* Denote $p_i = \mathbb{P}(Y = i)$ as the clean label distribution, $\tilde{p}_i = \mathbb{P}(\widetilde{Y} = i)$ as the clean label distribution. Let $\epsilon' = \frac{K \cdot \epsilon}{K-1}$, we have:

$$\mathbb{E}_{(X,\widetilde{Y})\sim\widetilde{\mathcal{D}}}\Big[(1-r)\cdot\ell\big(\mathbf{f}(\mathbf{X}),\widetilde{Y}\big)\Big] + \mathbb{E}_X\Big[\sum_{i\in[K]}\frac{r}{K}\cdot\ell\big(\mathbf{f}(\mathbf{X}),i\big)\Big]$$

$$=\Big[\sum_{i\in[K]}\mathbb{E}_{(X,\widetilde{Y})\sim\widetilde{\mathcal{D}},Y=i}\Big[(1-r)\cdot\ell\big(\mathbf{f}(\mathbf{X}),\widetilde{Y}\big)\Big]\Big] + \mathbb{E}_X\Big[\sum_{i\in[K]}\frac{r}{K}\cdot\ell\big(\mathbf{f}(\mathbf{X}),i\big)\Big]$$

$$=\Big[(1-r)\cdot\sum_{i\in[K]}\mathbb{E}_{(X,\widetilde{Y})\sim\widetilde{\mathcal{D}},Y=i}\Big[\sum_{j\in[K]}T_{i,j}\cdot\ell\big(\mathbf{f}(\mathbf{X}),\widetilde{Y}=j\big)\Big]\Big] + \mathbb{E}_X\Big[\sum_{i\in[K]}\frac{r}{K}\cdot\ell\big(\mathbf{f}(\mathbf{X}),i\big)\Big]$$

$$=\Big[(1-r)\cdot\sum_{i\in[K]}\mathbb{E}_{X,Y=i}\Big[(1-\epsilon')\cdot\ell\big(\mathbf{f}(\mathbf{X}),i\big) + \sum_{j\in[K]}\frac{\epsilon'}{K}\cdot\ell\big(\mathbf{f}(\mathbf{X}),j\big)\Big]\Big] + \mathbb{E}_X\Big[\sum_{i\in[K]}\frac{r}{K}\cdot\ell\big(\mathbf{f}(\mathbf{X}),i\big)\Big]$$

$$=\Big[(1-r)\cdot\sum_{i\in[K]}\mathbb{E}_{X,Y=i}\Big[\big(1-\epsilon'\big)\cdot\ell\big(\mathbf{f}(\mathbf{X}),i\big)\Big]\Big] + \mathbb{E}_X\Big[\Big[\frac{(1-r)\cdot\epsilon'}{K} + \frac{r}{K}\Big]\sum_{j\in[K]}\ell\big(\mathbf{f}(\mathbf{X}),j\big)\Big]$$

$$=\Big[\underbrace{(1-r)\cdot\big(1-\epsilon'\big)}_{:=c_3}\mathbb{E}_{(X,Y)\sim\mathcal{D}}\Big[\ell\big(\mathbf{f}(\mathbf{X}),Y\big)\Big]\Big] + \mathbb{E}_X\Big[\underbrace{\Big[\frac{(1-r)\cdot\epsilon'}{K} + \frac{r}{K}\Big]}_{:=c_4}\sum_{j\in[K]}\ell\big(\mathbf{f}(\mathbf{X}),j\big)\Big]$$

$$=\Big[\frac{c_3}{1-r^*}\cdot\mathbb{E}_{(X,Y)\sim\mathcal{D}}\Big[\ell\big(\mathbf{f}(\mathbf{X}),Y^*\big) - \frac{r^*}{K}\sum_{j\in[K]}\ell\big(\mathbf{f}(\mathbf{X}),j\big)\Big]\Big] + \Big[c_4\cdot\mathbb{E}_X\Big[\sum_{j\in[K]}\ell\big(\mathbf{f}(\mathbf{X}),j\big)\Big]\Big]$$

$$=\underbrace{\Big[\frac{c_3}{1-r^*}\cdot\mathbb{E}_{(X,Y)\sim\mathcal{D}}\Big[\ell\big(\mathbf{f}(\mathbf{X}),Y^*\big)\Big]\Big]}_{\text{True Risk}} + \underbrace{\Big[\Big(c_4 - \frac{c_3\cdot r^*}{(1-r^*)\cdot K}\Big)\cdot\mathbb{E}_X\Big[\sum_{j\in[K]}\ell\big(\mathbf{f}(\mathbf{X}),j\big)\Big]\Big]}_{\text{M-Inc1}}$$

Adopting $r_{\text{opt}} = \frac{r^*-\epsilon'}{1-\epsilon'}$, with a bit of math, the weight of Term M-Inc1 becomes 0 and

$$\mathbb{E}_{(X,\widetilde{Y})\sim\widetilde{\mathcal{D}}}\Big[\ell\big(\mathbf{f}(\mathbf{X}),Y^{\text{GLS},r_{\text{opt}}}\big)\Big]$$

$$=\mathbb{E}_{(X,\widetilde{Y})\sim\widetilde{\mathcal{D}}}\Big[(1-r_{\text{opt}})\cdot\ell\big(\mathbf{f}(\mathbf{X}),\widetilde{Y}\big)\Big] + \mathbb{E}_X\Big[\sum_{i\in[K]}\frac{r_{\text{opt}}}{K}\cdot\ell\big(\mathbf{f}(\mathbf{X}),i\big)\Big]$$

$$=\Big[\frac{c_3}{1-r^*}\cdot\mathbb{E}_{(X,Y)\sim\mathcal{D}}\Big[\ell\big(\mathbf{f}(\mathbf{X}),Y^*\big)\Big]\Big] \iff \mathbb{E}_{(X,Y)\sim\mathcal{D}}\Big[\ell\big(\mathbf{f}(\mathbf{X}),Y^*\big)\Big]$$

$\square$

