# OpenReview forum: "Understanding Generalized Label Smoothing when Learning with Noisy Labels"
_ICLR.cc/2022/Conference — ICLR 2022 Submitted_

### Official Review · Reviewer_Ww4B · 2021-11-01

**Correctness:** 3
**Technical Novelty And Significance:** 2
**Empirical Novelty And Significance:** 2
**Recommendation:** 5
**Confidence:** 3

**Main Review:**

This paper provides understandings for a generalized notion of label smoothing (GLS) when learning with noisy labels and theoretically show that negative label smoothing improves the expected model confidence over the data distribution. The empirical experiments on multiple benchmark datasets demonstrate that with the presence of label noise, negative label smoothing (NLS) becomes competitively robust to label noise.

However, to the best of my knowledge, the following points need to be addressed before it can be published:

1)	My first concern is the originality and novelty of this work. In this work distributed soft label learning is used. To my knowledge, label distribution learning is a related research topic about soft label-based multi-label learning, which should be discussed.
2)	For Figure 1, more detailed descriptions and explanations should be given.
3)	The last paragraph in Section Introduction, please don't cite so many references at the same place. I suggest to delete some unimportant references and then dispersedly cite these references.
4)	Two UCI datasets are used to evaluate the effectiveness of the proposed algorithm. My question is why the authors choose these datasets. To my knowledge, there are a large number of UCI datasets can be used for evaluating the effectiveness of the proposed algorithm. More datasets may be needed. At least, please explain the reason why these datasets are used in the current experiments.
5)	It is more convinced if the authors can compare the proposed algorithm with some other state-of-the-art label distribution learning algorithms.
6)	With regard to the comparison results, statistical tests are needed in the comparison results.
7)	The presentation can be further polished. For example, some equations have punctuation at the end and some don't have.


**Summary Of The Paper:**

This paper provides understandings for a generalized notion of label smoothing (GLS) when learning with noisy labels and theoretically show that negative label smoothing improves the expected model confidence over the data distribution.

**Summary Of The Review:**

This paper provides understandings for a generalized notion of label smoothing (GLS) when learning with noisy labels and theoretically show that negative label smoothing improves the expected model confidence over the data distribution. The empirical experiments on multiple benchmark datasets demonstrate that with the presence of label noise, negative label smoothing (NLS) becomes competitively robust to label noise.

However, to the best of my knowledge, the following points need to be addressed before it can be published:

1)	My first concern is the originality and novelty of this work. In this work distributed soft label learning is used. To my knowledge, label distribution learning is a related research topic about soft label-based multi-label learning, which should be discussed.
2)	For Figure 1, more detailed descriptions and explanations should be given.
3)	The last paragraph in Section Introduction, please don't cite so many references at the same place. I suggest to delete some unimportant references and then dispersedly cite these references.
4)	Two UCI datasets are used to evaluate the effectiveness of the proposed algorithm. My question is why the authors choose these datasets. To my knowledge, there are a large number of UCI datasets can be used for evaluating the effectiveness of the proposed algorithm. More datasets may be needed. At least, please explain the reason why these datasets are used in the current experiments.
5)	It is more convinced if the authors can compare the proposed algorithm with some other state-of-the-art label distribution learning algorithms.
6)	With regard to the comparison results, statistical tests are needed in the comparison results.
7)	The presentation can be further polished. For example, some equations have punctuation at the end and some don't have.

---

> ### Author Response · Authors · 2021-11-22
> **Response to Reviewer Ww4B**
>
> $\textbf{Dear Reviewer Ww4B},$
>
> We would like to thank Reviewer Ww4B for reviewing our paper and providing suggestions. Please see our response below:
>
> **1. Originality and novelty of our work, discussions about label distribution learning**
>
> **Response:**
>
>  We agree with Reviewer Ww4B that Label Smoothing can be viewed as a special kind of soft label, and we **included discussions about label distribution learning in the full version of related works** (Appendix A, highlighted in blue). However, **we indeed are not claiming that we provide an understanding of soft labels.** As described in [1], label distribution provides an instance with description degrees of all the labels, which can be viewed as an instance-dependent structural form of soft labels, mathematically, the label of feature x_i is denoted as $[d_{x_i}^{y_j}]$ for $j \in [K]$. While label smoothing becomes a popular topic is due to its **simplicity and effectiveness** in various tasks. Thus, label distribution learning differs significantly from label smoothing in its instance dependency as well as the non-uniform structure of soft labels. And we respectfully disagree with Reviewer Ww4B that we should compare with state-of-the-art label distribution learning methods to show the effectiveness of GLS. It seems that in Bullet 5, the reviewer misunderstood our work as a method paper. However, we did include the comparisons between GLS and other robust methods to test the effectiveness of GLS in practice:
>
> **Effectiveness of GLS in addressing label noise** To better explain the effectiveness of GLS, we moved comparisons between GLS and more robust methods from Appendix to Section 6.2. Please take a look!  We want to highlight that GLS (especially NLS) outperforms several robust methods and meanwhile maintains its simple form and is both effective and efficient (easy to implement) in practice.
>
> **Our novelty and contributions:** we also want to highlight that our novelty and contributions can be summarized as:
>
> -	**Theoretically:** we show several robust loss methods correspond to generalized label smoothing where the smooth rate can go negative with the presence of label noise.
> -	**We address a conflict:** positive label smooth mitigates label noise [2], but existing robust solutions seem to be closer to learning with negatively smoothed labels (NLS). Theoretically, we reveal that PLS vanishes in a high label-noise regime, and NLS then becomes more beneficial.
> -	**Empirically:** Label smoothing is popular due to its simplicity and effectiveness in various applications. From the aspect of model confidence as well as our empirical observations, we empirically show that when NLS fails to work, learning with CE and then applying a negative label smoothing regularizer significantly improves the model performance, and even outperforms the positive label smoothing. And we want to **highlight the importance of NLS**, which could be beneficial for the scenario where PLS fails to work, and meanwhile even has better performances than PLS in successful applications of positive label smoothing.
>
> **2. More descriptions of  Figure 1:**
>
> **Response:**
>
>  In Figure 1, we adopt various label smoothing rates $$r\in [-8, -4, -2, -1, -0.8, -0.6, -0.4, -0.2, 0.0, 0.2, 0.4, 0.6, 0.8]$$ and train on 3 UCI datasets. We visualize what smooth rates are best (top 2, possible to have tied smooth rates) under different noise levels (noise rate is 0.0, 0.1, 0.2, 0.3, 0.4). Clearly, with the increasing of the noise rates, we can see an overall decreasing trend in the optimal smooth rate: the best smooth rate (red) tends to be positive when the noise rate is low. While it is beneficial to use a negative smooth rate (green) when the noise rate is high. The experiment settings of the UCI dataset is available in Appendix D.3. And experiment results (test accuracies, etc) are deferred to Table 1 and Table 8 (Appendix).
>
> Our main purpose for Figure 1 is to share our observation that label smoothing becomes less effective when the label noise rate is high, which complements [2] that label smoothing is beneficial when learning with noisy labels.
>
> **3. Dispersedly cite references:**
>
> **Response:**
>
>  Thanks for the suggestion, we have revised the last paragraph in the **Introduction**. And we colored the differences in **blue** in our revised version.
>
> References:
>
> [1] Label distribution learning, IEEE Transactions on Knowledge and Data Engineering 28.7 (2016)
>
> [2] Does label smoothing mitigate label noise? ICML 2020.

---

> > ### Author Response · Authors · 2021-11-22
> > **Additional Response to Reviewer Ww4B**
> >
> > **3. Why do we use UCI  datasets?**
> >
> > **Response:**
> >  Our UCI dataset selections follow from two popular works [ 3, 4]. And we did include 4 more UCI datasets, synthetic 2D and synthetic CIFAR datasets in our initial submission. (Please refer to Appendix D.4 for experiments on synthetic datasets and more UCI datasets, also Section 6 for synthetic CIFAR datasets)
> >
> > **4. Statistical tests are needed in the comparison results**
> >
> > **Response:**
> >
> >  Thanks Reviewer Ww4B for the suggestion. We adopt the paired t-test to perform the statistical test. Given the noise rate, group 1 consists of $5\times4$ test accuracies of PLS on the test data (top 5 test accuracies for each smooth rate in $[0.2, 0.4, 0.6, 0.8]$). Group 2 consists of $5\times4$ test accuracies of NLS on the test data (top 5 test accuracies for each smooth rate in $[-2.0, -4.0, -6.0, -8.0]$). With the paired t-test, we show whether switching the positive smooth rate to a negative one can be beneficial in improving the model performance (test accuracy) under each noise rate.
> > We adopted the CIFAR-10 symmetric noise as an example (PLS vs NLS):
> >
> > |Noise rate|t-value|p-value|
> > |:-----:|:-----:|:-----:|
> > | 0.2 |  -3.57  |  0.0020  |
> > | 0.4 |   -20.32   |   2.40e-14 |
> > | 0.6 |   -11.44    |   5.82e-10 |
> >
> > Thus, we can safely conclude that NLS generally outperforms PLS with a p-value for each noise rate being < 0.05. The negative difference is statistically significant. Besides, the p-value of 0.4, 0.6 noise rates is much lower than the low noise scenario (0.2).
> >
> > **5. Polishing the presentation**
> >
> > **Response:**
> >
> > Thanks for pointing out this issue. We have removed the punctuations in all equations.
> >
> > References:
> >
> > [3] Learning with noisy labels, NeurIPS 2013.
> >
> > [4] Peer loss functions: Learning from noisy labels without knowing noise rates, ICML 2020.

---

### Official Review · Reviewer_2zXX · 2021-11-02

**Correctness:** 4
**Technical Novelty And Significance:** 3
**Empirical Novelty And Significance:** 4
**Recommendation:** 5
**Confidence:** 4

**Main Review:**

Strength:

The paper is very well motivated, and revealed new findings for the power of label smoothing. The paper provided empirical instructions for when a negative or a positive smoothing rate should be adopted.

The proposed generalized label smoothing scheme ties closely to several existing learning with noisy labels solutions. This new connection might reveal a deep connection between label smoothing and other regularization techniques in the literature of learning with noisy labels. This seems to provide a set of new understandings to the literature of learning with noisy labels.

The discovered phase transitioning behavior is supported both theoretically and empirically. This result quantifies the scenarios of using positive and negative label smoothing.

The presented experiment results are thorough and complete. The analysis is well presented.

Weakness & question:

The discussion in Section 4 on the bias-variance tradeoff seems to disconnect a bit with Section 5. Could the authors clarify its use?

The characterization of the phase transitioning behavior seems to depend on the knowledge of r^*, the optimal smooth rate when data is clean. This knowledge is important to inform when one should adopt a negative smoothing rate and when a positive one. What will happen when only an imperfect r^* is known?



**Summary Of The Paper:**

Label smoothing shows a promising bias-variance tradeoff when training deep neural networks, while it is unclear if label smoothing remains helpful when training labels are noisy. A recent work shows the benefits of using label smoothing. However, the current submission finds contradicting and appealing observations. In Figure 1, the authors presented a convincing set of results that clearly show a phase-transitioning behavior of using label smoothing: while label smoothing seems to work with a low label noise rate, the benefits disappear when the rate goes up high.

The above observation can imply that label smoothing is considered unfavorable with a high label noise rate. The paper then presented a generalized label smoothing scheme where the smoothing constant can instead go negative. Even though using a negative parameter sounds weird, the paper showed that a label smoothing with a negative (but different) smoothing parameter corresponds to several existing solutions to learning with noisy labels! This signals that the negative label smoothing scheme might turn out to be helpful when dealing with high label noise rate. I think this is a pretty interesting observation and connection to uncover.

The two cases are integrated as the generalized label smoothing, where the smoothing constant can be either negative to positive. In section 5, it is shown under what noise rate, a positive or a negative label smoothing is considered a better option. This characterization reveals a clear phase transitioning behavior corresponding to figure 1.

**Summary Of The Review:**

I tend to accept this paper. It does explain some interesting things, and the experiment is also sufficient.

========== post rebuttal ============

The rebuttal of this article solves my concerns to some extent, but as other reviewers pointed out, some problems still exist, so I changed my score.

---

> ### Author Response · Authors · 2021-11-22
> **Response to Reviewer 2zXX**
>
> $\textbf{Dear Reviewer 2zXX,}$
>
>
> We would like to thank Reviewer 2zXX for reviewing our paper and providing suggestions. Please see our response below:
>
> **1. Section 4 disconnects a bit with Section 5**
>
> **Response:**
>
> We agree with Reviewer 2zXX that the discussion of the bias-variance trade-off somehow disconnects with Section 5.
> -	**Our initial purpose:** In our initial version, we aimed to show that by referring to the empirical bias and variance trade-off of GLS, we could naturally introduce the assumption 1 for the optimal smooth rate.
> -	**Our revised storyline:** In our revised version (differences are highlighted in blue), we revised Section 4.2 by introducing some insights of model confidence from empirical observations, i.e, GLS with non-positive smooth rates may yield over-confident model predictions and reduce the test accuracy. With the presence of label noise, model confidence inevitably degrades. A smaller smooth rate tends to be more beneficial. Based on these observations, we proceed to Section 5 and show when we should adopt NLS, and when not.
>
> **2. What if imperfect $r^{\*}$ is known? How to choose NLS and PLS in practice?**
>
> **Response:**
>
>  We did include the **discussion of practical considerations of GLS in Appendix C** in our initial submission:
> -	**Appendix C.1 includes how to select $r_{opt}$ in practice.** From an empirical aspect, one can simply assume $r^*\to 0$. And the noise rate $e$ is estimable by a large family of noise estimation methods such as [1, 2, etc]. Our practical observations show that NLS with a CE warm-up is not sensitive to the negative smooth rate, for example, on CIFAR-10 and CIFAR-100 synthetic noisy datasets, $r<-1.0$ frequently achieves the best results (please see Table 2 in the main paper). Our current contribution focuses on understanding the generalized label smoothing, and we prefer leaving the task of identifying the optimal smooth rate to future works.
> -	By adopting existing methods in noise rate estimation and noisy label detections, **we introduce how to practically make GLS more robust to label noise in Appendix C.2.**
> -	We also extended Theorem 5 to **multi-class** in order to improve the practicality (the optimal smooth rate in the multi-class classification task).
> -	To show the effectiveness of GLS in practice, we moved **comparisons between GLS and more robust methods** from Appendix to Section 6.2. Please take a look!  GLS (especially NLS) outperforms several robust methods and meanwhile maintains its simple form and is both effective and efficient (easy to implement) in practice.
>
> References:
>
> [1] Classification with Noisy Labels by Importance Reweighting, TPAMI 2015.
>
> [2] Learning from Corrupted Binary Labels via Class-probability Estimation, ICML 2015.

---

### Official Review · Reviewer_ENj2 · 2021-11-07

**Correctness:** 2
**Technical Novelty And Significance:** 1
**Empirical Novelty And Significance:** 1
**Recommendation:** 3
**Confidence:** 4

**Main Review:**

- The major problem with the paper is analyzing the expected loss. Label smoothing is a regularizer which has been shown to improve generalization and model calibration. However, in the infinite sample limit (minimizing the expected loss) it makes no sense to add any regularizer since directly minimizing the expected loss is optimal, e.g., In eq. 4 the authors analyze the expected loss (under the data distribution) subject to a label smoothing penalty. Why does this make sense ?
- The authors claim that negative label smoothing is different from standard label smoothing and present it as a significant result. However, this is trivial. The label-smoothed cross entropy loss can be equivalently written in the constrained form as:

    $$\min l(f(x),y) \quad s.t.~ H(U, f(x)) \leq c$$

    where U is the discrete uniform distribution and $l(.)$ is the log-likelihood. For negative label smoothing the inequality constrained is reversed. By virtue of the constraints, the minimizer of NLS objective will have greater expected margin (or model confidence as called in the paper) than standard LS which pushes the model predicted class distributions to be closer to uniform.

- The claim that negative label smoothing performs well in the high label noise regime is not surprising both from a theoretical and practical standpoint. The point of using label smoothing is to prevent the distribution learned by neural networks to become too peaked (low entropy or overconfident). But when the label noise is high, with labels flipped randomly, then the model trained without any label smoothing is not going to be overconfident (the expected loss of the model is at least the Bayes error and with label noise the Bayes error is already high).
- The paper doesn't position itself clearly with respect to related work. Chen et al 2020 have already studied generalization performance under label smoothing where, unlike this paper, they study the generalization performance of the finite sample minimizer under label smoothing and label noise. While that paper is cited, the results of this paper are not put in context with their results.

## Other comments

- In Theorem 3, what does it mean for the smoothing rate $r_{CL}$ to approach infinity ? In this case the likelihood term is completely ignored.
- High label noise is never clearly defined.

**Summary Of The Paper:**

The paper considers generalized label smoothing (GLS) for binary classification tasks where the smoothing coefficient is allowed to be negative (negative label smoothing, NLS). The authors show the connection between GLS and several existing loss functions through elementary algebraic manipulations of the expected loss under (generalized) label smoothing. They also analyze GLS in the noisy label setting where the labels are randomly flipped and show that using negative label smoothing is desirable in the high label noise setting.

**Summary Of The Review:**

The main problem with the paper is analyzing the effect of label smoothing on the minimizer of the expected loss of a classifier. There is no need for any regularization when minimizing the expected loss. This is a fundamental conceptual problem with the paper. Furthermore, the results are fairly elementary and do not significantly contribute towards theoretical understanding of label smoothing beyond existing literature (c.f. Chen et al 2020).

---

> ### Author Response · Authors · 2021-11-22
> **Response to Reviewer ENj2 (Part 1)**
>
> $\textbf{Dear Reviewer ENj2,}$
>
> We would like to thank Reviewer ENj2 for reviewing our paper and providing suggestions. Please see our response below:
>
> **1. ''Analyzing the expected loss makes no sense''**
>
> **Response:**
>
> We disagree with Reviewer ENj2 that **Analyzing the expected loss makes no sense**. We want to clarify that Reviewer ENj2 has the following misunderstandings of our work:
>
>
> - **Eqn. 4 includes the case of finite samples:** in equation 4, we are not restricting the data distribution to the infinite-sample. Suppose we are interested in the discrete distribution $\mathcal{D}$ such that $\mathcal{D}$ is the set of $(x_i, y_i)$, for $i \in [N]$. Equation 4 then becomes:
> $$\min \Big[\frac{1}{N}\sum_{i\in[N]} \ell(\mathbf{f}(\mathbf{x_i}), y_i)\Big] +\frac{ r}{2N} \Big[\sum_{i\in[N]}\big( \ell(\mathbf{f}(\mathbf{x_i}), 1-y_i)-\ell(\mathbf{f}(\mathbf{x_i}), y_i)\big)\Big]$$
>
> -	**$\mathcal{D}$ is not equal to $\mathcal{D}_{test}$:**   In assumption 1, we assume that the unseen test data distribution $\mathcal{D}_{test}$ may differ from $\mathcal{D}$, which is the clean (train) data distribution. This difference can be very well due to the training distribution coming from the finite number observation. In this case, minimizing the expected loss w.r.t hard labels may not yield the best performance on the test distribution.
>
> **2. ''The difference between LS and NLS is trivial''**
>
> **Response:**
>
> This seems to be another misunderstanding. **Our major claim of contribution** is understanding label smoothing under label noise and connecting negative label smoothing to existing learning with noisy label techniques  (Theorem 2, 3, 4 in Section 3). **Unfortunately, Reviewer ENj2 seems to have ignored our contributions and only focuses on the proposal of negative label smoothing.**
> Clearly, if we only consider the formulation, their differences lie in the sign of the weight of the uniform label. However, we are not proposing NLS methods, instead,
> -	We highlight the importance of NLS in the role of model confidence as well as its robustness with the presence of noisy labels. As an illustration, we adopt the learning-with-noisy-label task as an application. We provide the understanding and new insights for why and when NLS may be more beneficial. To our best knowledge, we did not observe existing evidence/works that claim negative label smoothing can do better than positive label smoothing in any applications or specific tasks.
> -	Meanwhile, we theoretically connect GLS with several popular robust loss functions and address the seemingly conflicts: why both NLS and PLS are robust, though it seems in the opposite views, from a theoretical aspect (Theorem 6 and Theorem 7: NLS is beneficial in high noise regime), and empirical aspect (Section 6 and Appendix D2: especially in Appendix D2, NLS with a CE warm-up significantly improves the performance of NLS, similarly for PLS in the high noise regime). This simple while effective empirical trick has great potential in fixing failed applications of PLS and further improvements of PLS.
>
> We respectfully disagree with Reviewer ENj2 that all above-mentioned contributions can be explained by the mentioned constrained form, especially:
>
> - why both NLS and PLS are robust (the seemingly conflicts);
> - the equivalent form of GLS and other popular robust loss functions (Theorem 2, 3, 4);
> - how to improve the performance of NLS on clean and synthetic noisy CIFAR datasets.
>
> **3. ''The claim that negative label smoothing performs well in the high label noise regime is not surprising both from a theoretical and practical standpoint''**
>
> **Response:**
>
> Unfortunately, we found **no formally documented evidence** of the Reviewer ENj2’s claim. If Reviewer ENj2 has a pointer to the literature, we’d be eager to learn. In Particular, **we found no theoretical justification**. Empirically, there were even **contradicting results** reported in [1]. Therefore it is unclear to us why the reviewers believed this is a trivial consequence.
>
> In particular, to support Reviewer ENj2’s claim on “negative label smoothing performs well in the high label noise regime is not surprising”, **we’d be grateful if Reviewer ENj2 can point us documented evidence for:**
> -	Theoretical evidence for why NLS performs well in the high label noise regime.
> -	Empirical evidence for NLS performs well in high label noise regime, and also in clean datasets such as CIFAR 10 or other popular datasets.
> -	Empirical and theoretical evidence for why both PLS and NLS could be robust to label noise.
>
> Reference
>
> [1] Does label smoothing mitigate label noise? ICML 2020.

---

> > ### Author Response · Authors · 2021-11-22
> > **Response to Reviewer ENj2 (Part 2)**
> >
> > **4. Chen et al (2020) [R2] have already studied generalization performance under label smoothing**
> >
> > [R2] An Investigation of how Label Smoothing Affects Generalization, ArXiv.
> >
> > **Response:**
> >
> > **We disagree with the reviewer’s claim that our results are fairly elementary and do not significantly contribute towards a theoretical understanding of label smoothing beyond [R2].**  Our reasons come as follows:
> >
> > **Differences between our work and [R2]:**
> >
> > -	**Nothing about NLS in [R2]:**  Unfortunately, [R2] focuses solely on the positive label smoothing, while our major contribution is the theoretical and empirical demonstration of negative label smoothing, as well as the understanding of the generalized notion of label smoothing. Clearly, whether, why, and when NLS will work are not discussed in [R2], neither in a theoretical view nor an empirical view.
> > -	**Theoretical difference 1: connection to other robust methods (our section 3)**  [R2] makes no theoretical conclusions w.r.t the connection between label smoothing and robust loss methods (such as backward/forward loss correction, complementary labels, peer loss functions, etc) when learning with noisy train labels under clean test data (so-called alpha-type theory in [R2]).
> > -	**Theoretical difference 2: the formulation differs significantly**  Reviewer ENj2 did not get the significant differences in formulation between our work (section 4 and 5) and [R2]: (1) **[R2] assumes that the distribution of two classes is balanced under finite samples ($\mathbf{P}(Y=1)=\mathbf{P}(Y=0)=0.5$), while we don’t have such a strong restriction**, and the data distribution in our setting can be either finite or infinite. Besides, we feel like binary classification and the same clean prior have **overly simplified** the theoretical study of label smoothing. (2) Our theoretical observations in Sections 4 and 5 build on Assumption 1 which is about the optimal smooth rate under the clean data. And we aim to theoretically demonstrate the transition of optimal smooth rate in our empirical observations.
> >  -	**Experiments:** [R2] has no experiments. While in our work, we provide extensive experiments on synthetic data, UCI datasets, synthetic CIFAR-10, and synthetic CIFR-100 to validate our theoretical conclusions and reveal new insights on making use of GLS (especially NLS).
> >
> > What is more, as we have mentioned in the third response, we are eager to study empirical/theoretical evidence that could show all our contributions are trivial.
> >
> > **5. The meaning of $r_{CL}\to -\infty$ in Theorem 3**
> >
> > **Response:**
> >
> > The likelihood term is not ignored. Mathematically, when $r_{\text{CL}}\to -\infty$, GLS with $r_{\text{CL}}$ has the following equivalent form (denote $r_{\text{CL}}$ as $r_c$ below):
> >
> > $$\mathbf{E}_{(X, \widetilde{Y})\sim \widetilde{\mathcal{D}}}\Big[ (1-\frac{r_c}{2}) \cdot \ell(\mathbf{f(X)}, \widetilde{Y})+\dfrac{r_c}{2}\cdot \ell(\mathbf{f(X)}, 1-\widetilde{Y})\Big]$$
> >
> > Which is further equivalent to:
> >
> > $$\mathbb{E}_{(X, \widetilde{Y})\sim \widetilde{\mathcal{D}}}[\ell(\mathbf{f(X)}, \widetilde{Y})+\dfrac{r_c}{2-r_c}\cdot \ell(\mathbf{f(X)}, 1-\widetilde{Y})$$
> >
> > And when $r_c\to -\infty$, we have:
> >
> > $$\mathbb{E}_{(X, \widetilde{Y})\sim \widetilde{\mathcal{D}}}[\ell(\mathbf{f(X)}, \widetilde{Y})- \ell(\mathbf{f(X)}, 1-\widetilde{Y})$$
> >
> > **5. High label noise is never clearly defined**
> >
> > **Response:**
> >
> >  Whether the noise rate is high or not depends on the number of classes, for example, 0.3 noise rate in binary task is already a high-level noise rate; while in the 10-class task, the noise rate of 0.3 is not that large. Besides, as shown in Figure 2, depending on the value of $r^*$ (characterizes the differences/dis-similarities among classes), the division of high and low noise rates can be decided by the boundary of red and green zones. Mathematically, $e=\frac{(K-1)r^*}{K}$ in Theorem 7 (multi-class, Appendix C).

---

> > ### Comment · Reviewer_ENj2 · 2021-11-22
> > **Post-rebuttal comments**
> >
> > I thank the authors for their detailed response. I did a quick comparison of the new submission with the original submission and it appears that the authors have significantly changed their paper. There is a new section on "insights from empirical observation" and crucially the authors have deleted Theorem 5 in section 4. Theorem 5 in the new submission is Theorem 6 in the original submission. I have re-produced the original Theorem 5 below:
> >
> > **Theorem 5.**  If $f^*_{r_N} \neq f^*_{r_P}$, we have  $E_{(X,Y) \sim \mathcal{D}}[MC(f^*_{r_N}; X, Y)] > E_{(X,Y) \sim \mathcal{D}}[MC(f^*_{r_P}; X, Y)]$.
> >
> > where $f^*_{r_N}$ (respectively $f^*_{r_P}$) is the minimizer of the *expected* label smoothing loss (wrt to distribution $\mathcal{D}$) with negative (respectively positive) label smoothing penalty $r$.
> >
> > In the above, setting $\mathcal{D}$ to be the empirical distribution over the $N$ points doesn't make sense since Theorem 5 will also be  with respect to the same empirical distribution $\mathcal{D}$. Theorem 5 wasn't about generalization at all, either from one distribution $\mathcal{D}$ to another $\mathcal{D}_{test}$ or from finite sample to expected loss.
> >
> > Now the authors have surreptitiously deleted the offending theorem and claiming that I misunderstood the paper. The authors are not forthcoming in this change that they have made to the paper. This seriously questions the effort that I have put into reviewing the paper and my intelligence. In light of this situation I am not willing to spend more time reviewing the updated paper unless the ACs feel that I am wrong in refusing to review the new changes. As such I strongly recommend rejection of the paper.

---

> > > ### Author Response · Authors · 2021-11-23
> > > **This seems to be another misunderstanding: we restructured Sec 4 based on the other two reviewers' comments**
> > >
> > > $\textbf{Dear Reviewer ENj2,}$
> > >
> > > This seems to be another misunderstanding. We restructured the original Section 4 and removed the results because:
> > >
> > > - **Theorem 5 was not our main result:** it was simply an observation. Reviewer 2zXX also thinks there exist disconnections between Section 4 and 5. So we decided to revise Section 4 (replace Theorem 5 with empirical observations to connect Section 4 closer to Section 5. )
> > >
> > > - **Reviewer nStq questioned its clarification so we decided to take it out and use more intuitive empirical observations.**
> > >
> > > We never hide this fact (all revision histories are indeed visible) and were **pretty upfront about it in our responses to the other two reviewers** (Reviewer nStq and Reviewer 2zXX). We didn't mention it when replying to reviewer ENj2 because we never thought Reviewer ENj2 implied this is the "offending theorem", for two reasons:
> > >
> > > 1. **We really didn't think Theorem 5 was relevant in this discussion.**
> > >
> > > 2. **Reproducing reviewer ENj2's initial comment:** "However, in the infinite sample limit (minimizing the expected loss) it makes no sense to add any regularizer since directly minimizing the expected loss is optimal, e.g., In eq. 4 the authors analyze the expected loss (under the data distribution) subject to a label smoothing penalty. ”
> > >
> > > No where we found the mention of Theorem 5 . All the "offending" Eqn. 3, 4, 5 (**the minimization of the expected loss**) are in Section 3 and Section 5, and we thought it was about our main results in Section 3 and 5, particularly Theorem 2, 3, 4 (Section 3) Theorems 6, 7 (Section 5) in our initial version. We made no modifications to all other theorems in our revised submission.
> > >
> > > Besides, Theorem 5 (in the initial submission) is about the model confidence of the optimal model for NLS and PLS, not about the optimization itself, nor relevant with the difference between $\mathcal{D}$ and $\mathcal{D}_{test}$.
> > >
> > > **We also didn’t think revising the paper substantially was an issue.** We thought this is exactly the purpose of using open review, which promotes a healthier development cycle of research results.
> > >
> > > We hope the reviewer does not jump too quickly to conclude that we ignored his/her effort, nor questioned his/her intelligence (no way we intended to do that). We in fact **appreciated** that the review comments helped us realize how the earlier presentation from us might have confused readers, and we believe we have improved the clarification, for that, we thank the reviewer.
> > >
> > > **If Reviewer ENj2 decides to not engage with the paper anymore, we understand. We are not super concerned about the rejection recommendation, but we were eager to hear further from the reviewer about the existing evidence that let the reviewer believe our results were trivial.**
> > >
> > > Sincerely
> > >
> > > ICLR 2022 Conference Paper500 Authors

---

### Official Review · Reviewer_nStq · 2021-11-08

**Correctness:** 3
**Technical Novelty And Significance:** 2
**Empirical Novelty And Significance:** 1
**Recommendation:** 5
**Confidence:** 5

**Main Review:**

The authors proposed a very simple yet interesting idea NLS. They theoretically shown NLS can improve the expected model confidence. When label noise rates are high, they shown NLS can be more robust to label noise compared with PLS. Then they demonstrated the theoretical findings by conducting experiments on UCI and CIFAR datasets. The paper is easy to understand and well-organized. It is very good to provide a section about how the proposed GLS connects to other robust methods. However, I have several questions/comments that need the authors to address.

(a). Since the authors didn't provide any experiments to compare GLS with other loss adjusted methods of learning with noisy label such as [1,2], we have no idea about the practicality of GLS and its practical performance when learning with noisy label. It would be better if the authors could compare GLS with other methods such as [1,2]. Otherwise, its practicality is limited.

(b). Although the authors have shown theoretical results, the unknown parameters (for example, $r^*$ and $e$ in Theorem 7) makes the proposed GLD less practical. That is to say, in general, we don't know how to use GLS in practice.

(c). In the proof of theorem 5 (on page 28), why the inequality (see below) after "Besides, we have ..." holds? It is not obvious to me.
\begin{equation*}
\mathbb{E}   \left[\log\left(\frac{f^*_r(X)_Y}{1-f^*_r(X)_Y}\right)^{r/2} - \log\left(f^*_r(X)_Y\right)\right] \\
\ge \mathbb{E} \left[\log\left(\frac{f^*_s(X)_Y}{1-f^*_s(X)_Y}\right)^{|r|/2} - \log\left(f^*_s(X)_Y\right)\right]
\end{equation*}

where $\mathbb{E}[\cdot] := \mathbb{E}_{(X, Y)\sim \mathcal D}[\cdot], r:=r_N, s:=r_P$

(d). Is the quantity model confidence commonly used in the literature? If so, please add a reference.

Reference

1. Symmetric cross entropy for robust learning with noisy labels, ICCV 2019.

2. Early-Learning Regularization Prevents Memorization of Noisy Labels, NeurIPS 2020.

========== comments after rebuttal ============

I would like to thank the authors for their response, which solves my concerns except for the proof of Theorem 5 (of the initial version). The result of original Theorem 5 is important to the contribution of this paper, which unfortunately was removed in current version. In addition, after reading other reviewers' comments, there are still some other problems. Therefore, I will keep my overall score.

**Summary Of The Paper:**

This paper studies label smoothing when learning with noisy label. It proposed generalized label smoothing (GLS) containing positive label smoothing (PLS) and negative label smoothing (NLS), where NLS allows the smoothing parameter to be negative. The authors found that when the noisy rate is high, NLS is more beneficial than PLS, both theoretically and empirically.

**Summary Of The Review:**

This is a theoretical understanding paper for lable smoothing when learning with noisy label. However, its practicality is limited.

---

> ### Author Response · Authors · 2021-11-22
> **Response to Reviewer nStq**
>
> $\textbf{Dear Reviewer nStq,}$
>
> We would like to thank Reviewer nStq for reviewing our paper and providing suggestions. Please see our responses below:
>
> **1. No comparisons with loss adjusted methods**
>
> **Response:**
>
> We definitely agree with Reviewer nStq that comparing with more loss-adjusted methods can help with illustrating the effectiveness of GLS in practice. Due to space limits, we previously deferred the result comparisons between GLS and other robust methods in the Appendix. In our revised version, **we move the empirical comparisons to Section 6.2 and include the experiment of SCE [1] and ELR [2] as well** (Please take a look at Table 3 in the revised paper). Results in Table 3 show that (generalized) label smoothing is robust to label noise in practice.
>
> **2. Parameters $r^{*}$ and $e$ are unknown**
>
> **Response:**
>
> This is a great question! We agree with Reviewer nStq that finding the best smooth rate is non-trivial in practice. **We did include the discussion of practical considerations of GLS in Appendix C.** We briefly discussed below:
>
> -	To estimate $r_{\text{opt}}=\frac{r^*-2e}{1-2e}$, one can simply assume $r^*\to 0$. And the noise rate $e$ is estimable by a large family of noise estimation methods such as [3, 4, etc]. Our practical observations show that NLS with a CE warm-up is not sensitive to the negative smooth rate, for example, on CIFAR-10 and CIFAR-100 synthetic noisy datasets, $r<-1.0$ frequently achieves best results (see Table 2 in the main paper). Our current contribution focuses on understanding the generalized label smoothing, and we prefer to leave the task of identifying the optimal smooth rate to future works.
>
> -	In Appendix C, we also introduce insights to make GLS more robust to label noise in practice, for example, in the asymmetric label noise pattern, multi-class extensions. Please take a look if you are interested!
>
> **3. More details for the proof of Theorem 5**
>
> **Response:**
>
> Sorry for the confusion. We omitted to introduce the condition while completing the proof of Theorem 5 in our initial submission. And we follow the suggestions from Reviewer 2zXX that Section 4 somehow disconnects with Section 5. In order to
> -	Make the connection between Section 4 and 5 closer;
> -	Clarify a misunderstanding from Reviewer ENj2;
> -	Include empirical comparisons between GLS and several robust methods;
>
> In our revised version, we remove Theorem 5 in the initial submission and include empirical insights (section 4.2) about the role of model confidence in understanding generalized label smoothing. We also moved the comparisons between GLS and more robust methods to Section 6.2 to validate the effectiveness of GLS.
>
> **4. The quantity of model confidence**
>
> **Response:**
>
> One classical notion of the model prediction confidence is defined as $p(f(x)=y)$ where $f$ is the model, $(x, y)$ denotes the data sample [5]. Our defined quantity **model confidence** is beneficial in understanding generalized label smoothing. It is close to the measure of adaboosting confidence defined in [6] which adopted the **margin** quantity. For the binary classification task, one can view the **model confidence** (in our work) as an affine transformation of the **model prediction confidence** (in [5]), mathematically,
> $$MC(f; x, y)=p(f(x)=y)-p(f(x)=1-y)=2p(f(x)=y)-1$$
>
> References:
>
> [1] Symmetric Cross Entropy for Robust Learning with Noisy Labels, ICCV 2019.
>
> [2] Early-Learning Regularization Prevents Memorization of Noisy Labels, NeurIPS 2020.
>
> [3] Classification with Noisy Labels by Importance Reweighting, TPAMI 2015.
>
> [4] Learning from Corrupted Binary Labels via Class-probability Estimation, ICML 2015.
>
> [5] On Calibration of Modern Neural Networks, ICML 2021
>
> [6] Explaining-adaboost, Empirical inference 2013.

---

### Author Response · Authors · 2021-11-22
**Rebuttal revision of our paper is available.**

$\textbf{Dear Reviewers and readers,}$

We want to thank the reviewers for their constructive suggestions! The differences between the revised version and the first version are highlighted in Blue. To summarize,

-	We revised the discussion of related works at the end of the Introduction section (papers are cited dispersedly).  We also include the discussion of label distribution learning in Appendix A, the full version of related works. [Thanks **Reviewer Ww4B** for these suggestions!]
-	We added additional clarification for Equation 4 in Section 4.1 to address one misunderstanding from **Reviewer ENj2**. [Sorry for the confusion!]
-	We revised Section 4.2 by introducing some insights of model confidence from empirical results. Thus, proceeding from Section 4 to Section 5 appears to be more fluent. [Thanks **Reviewer 2zXX** for the suggestions!]
-	We moved the empirical comparisons to Section 6.2 and include the experiment of SCE [1] and ELR [2] as well (Please take a look at Table 3 in the revised paper). Results in Table 3 show that (generalized) label smoothing is robust to label noise in practice. Although we still want to highlight that we aim to provide new understandings of label smoothing, rather than requiring state-of-the-art performance by referring to the simple form of label smoothing. [Thanks **Reviewer nStq** for the suggestions!]

Sincerely

ICLR 2022 Conference Paper500 Authors

---

### Decision · Program_Chairs · 2022-01-20

**Decision:**

Reject

**Comment:**

This paper studies generalized label smoothing (GLS), which unifies positive label smoothing (PLS) and negative label smoothing (NLS), and studies its connections to existing loss functions. It also shows the benefit of NLS in the high noise regime.

Although the reviewers acknowledge that the idea of NLS in this paper is interesting, they also expressed the concerns that: the practicality of GLS is not thoroughly evaluated against prior works; the empirically best setting of parameter r is only verified in a limited number of datasets; the theoretical results' difference with prior works is limited. We encourage the authors to take the reviewers' feedback to strengthen the paper in the next iteration.